# Live-cell quantitative monitoring reveals distinct, high-affinity Gβγ regulations of GIRK2 and GIRK1/2 channels

Reem Handklo-Jamal [1], Tal Keren Raifman[1], Boris Shalomov [1], Patrick Hofer [2], Uri Kahanovitch[1], Theres Friesacher [2], Galit Tabak[1], Vladimir Tsemakhovich[1], Haritha P. Reddy [1,9], Orna Chomsky-Hecht[3], Debi Ranjan Tripathy[1,3], Kerstin Zuhlke[4], Carmen W. Dessauer [5], Enno Klussmann [4,6], Yoni Haitin [1,7], Joel A. Hirsch [3,7], Anna Stary-Weinzinger [2] ✉, Daniel Yakubovich [8] ✉ & Nathan Dascal [1,7] ✉

$G_{i/o}$ protein-coupled receptors (GPCRs) inhibit cardiac and neuronal excitability via G protein-activated $K^+$ channels (GIRK), assembled by combinations of GIRK1 - GIRK4 subunits. GIRKs are activated by direct binding of the Gβγ dimer of inhibitory $G_{i/o}$ proteins. However, key aspects of this textbook signaling pathway remain debated. Recent studies suggested no $G_{i/o}$-GIRK pre-coupling and low (>250 μM) Gβγ-GIRK interaction affinity, contradicting earlier sub-μM estimates and implying low signaling efficiency. We show that Gγ prenylation, which mediates Gβγ membrane attachment required for GIRK activation, also contributes to the Gβγ-GIRK interaction, explaining the poor affinity obtained with non-prenylated Gβγ. Using quantitative protein titration and electrophysiology in live *Xenopus* oocytes, Gβγ affinity for homotetrameric GIRK2 ranges from 4-30 μM. Heterotetrameric GIRK1/2 shows a higher Gβγ apparent affinity due to the Gβγ-docking site (anchor) in GIRK1, which enriches Gβγ at the channel. Biochemical approaches and molecular dynamic simulations reveal that the Gβγ anchor is formed by interacting N-terminal and distal C-terminal domains of the GIRK1 subunits, distinct from the Gβγ-binding "activation" site(s) underlying channel opening. Thus, the affinity of Gβγ-GIRK interaction is within the expected physiological range, while dynamic pre-coupling of Gβγ to GIRK1-containing channels through high-affinity interactions further enhances the GPCR-$G_{i/o}$-GIRK signaling efficiency.

G protein-activated inwardly rectifying $K^+$ channels (GIRK; Kir3) mediate inhibitory effects of $G_{i/o}$ protein-coupled receptors (GPCRs), controlling neuronal and cardiac excitability; GIRK malfunction is linked to neurological, cardiac and endocrine disorders[1–4]. GIRKs form homotetramers (GIRK2, GIRK4) or heterotetramers (GIRK1/2, 1/4, 1/3, 2/3), differing in tissue distribution and gating properties.

Homotetrameric GIRK2 is best characterized quantitatively, including a crystal structure of GIRK2-Gβγ complex[5]. GIRKs are activated by direct, cooperative binding of up to 4 molecules of Gβγ[6–9] (Fig. 1a). This membrane-delimited process requires posttranslational Gγ prenylation, essential for Gβγ accumulation at the plasma membrane (PM)[10] and GIRK activation[11,12].

**Fig. 1 | Lipid modification of Gγ is essential for GIRK activation and important for GIRK-Gβγ interaction. a** scheme of Gβγ activation of the GIRK2 channel. An agonist-bound GPCR (m2R) interacts with the $G\alpha_{i/o}\beta\gamma$ heterotrimer ($G\alpha_{i1}\beta_1\gamma_2$, PDB: 1gp2), catalyzing the GDP-GTP exchange at $G\alpha_{i/o}$ and its separation from Gβγ. Up to four Gβγ molecules bind sequentially to GIRK2. Channel opens when all four Gβγ-binding sites are occupied. The scheme shown represents the WTM model for the case of constant $PIP_2$ and $Na^+$ concentrations. **b** whole-cell currents in oocytes expressing GIRK2 and m2R without Gβγ (left), with Gβγ (middle), or with $Gβγ_{C68S}$ (right). Switching from a low-K to a high-K external solution (here 96 mM $[K^+]_{out}$) reveals $I_{basal}$. ACh (10 μM) elicits $I_{evoked}$, and then GIRK is blocked by 2.5 mM $Ba^{2+}$, revealing the non-GIRK background current. RNA doses (ng/oocyte) were: m2R, 1; GIRK2, 2; Gβ, 5; Gγ or $Gγ_{C68S}$, 2. **c, d** only Gβγ, but not $Gβγ_{C68S}$, increased $I_{basal}$ (**c**) and abolished $I_{evoked}$ (**d**). Boxes show the 25th–75th percentiles, whiskers indicate the minimum and maximum, and the line represents the median. Number of oocytes in each group is shown below the boxes (encircled numbers). Statistics: Kruskal-Wallis test with Dunn's multiple comparison vs. control (GIRK2 + m2R). One experiment, representative of two. **e** linear presentation of G1NC, G2NC and the truncated constructs. The transmembrane (TM) domains were replaced by a linker. **f** purified prenylated His-$Gβγ_{WT}$, captured on Ni-NTA beads, pulls down various [$^{35}$S] Met-labeled *ivt* proteins better than the non-prenylated $Gβγ_{C68S}$. *Top*, Coomassie staining of eluted proteins. Ni-NTA beads bound equal amounts of His-Gβγ and His-$Gβγ_{C68S}$. *Middle*, autoradiogram of a separate gel of 1/60$^{th}$ of the initial reaction mix (input). *Bottom*, autoradiogram of Gβγ-bound *ivt* proteins eluted from the beads (same gel as in upper image). Full gels are shown in Supplementary Fig. 2. **g** summary of binding to Gβγ of *ivt* proteins (% of input of the same protein). Bars show mean ± SEM; numbers of independent experiments for each protein are shown (encircled). Statistics for binding to His-Gβγ vs. His-$Gβγ_{C68S}$: unpaired t-test (Mann-Whitney test for G1NC).

The coupling between GPCR and $G\alpha_{i/o}\beta\gamma$ in the GPCR-$G\alpha_{i/o}\beta\gamma$-GIRK cascade varies by receptor, G protein and cell type, ranging from collision-coupling (e.g., muscarinic m2 receptor (m2R)[13–16]) to pre-coupling within dynamic multiprotein complexes (e.g., GABAB receptor with $G_{i/o}$ and GIRK[17]), or a combination of both modes within protein-enriched membrane "hot spots"[18], organized by specific scaffolds[19] or driven by low-affinity protein interactions[20].

Controversies linger regarding the affinity, specificity, and efficiency of $G\alpha_{i/o}\beta\gamma$-GIRK coupling. Early in vitro measurements of GIRK interaction with prenylated Gβγ yielded dissociation constants ($K_d$) between 50-800 nM[8,21], comparable to other Gβγ interactors (3 nM-3 μM; Supplementary Table 1). Contrastingly, an NMR study reported a $K_d$ of 250 μM for the interaction of non-prenylated Gβγ with GIRK1's truncated cytosolic domain[22]. Wang, Touhara, MacKinnon and colleagues analyzed Gβγ activation of purified recombinant GIRK2 while controlling Gβγ's surface density by titrating a non-prenylated His-Gγ into GIRK2- and NTA lipid-containing bilayers. Their studies revealed high cooperativity of Gβγ binding and its allosteric enhancement by $Na^+$ and $PIP_2$[15,23–25]. The resulting model, termed here WTM model, postulated sequential Gβγ binding to GIRK2, with channel opening when all four Gβγ sites are occupied[15,24] (Fig. 1a). Unexpectedly, binding of the first Gβγ showed an exceptionally low affinity, with $K_d \sim 1.9$ mM at $[Na^+]=0$ and ~300 μM at saturating $[Na^+]$[24]. (Due to cooperativity, the affinity increases for subsequent Gβγ bindings; Supplementary Table 2).

Low affinity entails inefficient signaling. With a $K_d > 250$ μM, GIRK activation (10-80%, depending on intracellular $Na^+$ concentration, $[Na^+]_{in}$) would require free surface Gβγ exceeding 1200 μm$^{-2}$

(molecules/$\mu m^2$)[24], hundredfold higher than the 2-10 $\mu m^{-2}$ GIRK channel density in PM of neurons or atrial myocytes[16,26]. While there is no evidence for such massive accumulation of Gαβγ around GIRKs, it could theoretically occur in membrane "hot spots". Alternatively, higher affinity or GIRK-G protein preassociation could enable fast and efficient signaling[27]. Several studies suggest preassociation of GIRKs with Gβγ or Gαβγ heterotrimers[17,28–32], while others support pure collision coupling[16,23,24]. Subunit-specific differences may play a role. GIRK1, but not GIRK2, recruits Gβγ to the PM; the cytosolic distal C terminal segment of GIRK1 (G1-dCT) is essential for Gβγ recruitment[33]. We previously proposed that G1-dCT is part of a Gβγ-docking site (Gβγ anchor) that facilitates high-affinity, dynamic (reversible) pre-association of GIRK1/2 with Gβγ[4,14,33–35]. However, the exact composition and interaction mode of the Gβγ-anchor remain unclear. Here we show that, besides driving Gβγ's attachment to the PM, Gγ's prenylation directly contributes to GIRK-Gβγ interaction. We quantitate interactions between Gβγ and GIRK channels in living cells by titrated protein expression and PM level monitoring, combined with biochemical assays and computational approaches. We demonstrate efficient, low-micromolar affinity, subunit-specific GIRK regulation by Gβγ and determine the composition of GIRK1's Gβγ-docking site. Our protein titration methodology can facilitate quantitative studies of additional membrane-delimited signaling cascades in living cells.

## Results

### Lipid modification of Gγ is essential for GIRK activation and important for GIRK-Gβγ interaction

All high-affinity estimates of Gβγ-GIRK binding were obtained using prenylated Gβγ. We hypothesized that Gγ's prenylation enhances Gβγ-GIRK interaction, similarly to Gβγ interactions with GPCRs, Gα, adenylyl cyclase and phospholipase Cβ[36–42].

In cells, the prenyl (geranylgeranyl in Gγ$_2$) moiety, Gγ$_{prenyl}$, is attached to Cys68 within the C-terminal CAAX motif, while the remaining residues are cleaved[10]. To assess the role of prenylation we used the non-prenylated mutant Gγ$_{C68S}$ that associates with Gβ[39]; however, Gβγ$_{C68S}$ fails to activate GIRK channels in excised PM patches[11,12]. We expressed GIRK2 channels with m2R (adjusted to maximize $I_{evoked}$[14]) and Gβ$_1$γ$_2$ (Gβγ) or Gβγ$_{C68S}$ in Xenopus oocytes and measured whole-cell basal ($I_{basal}$), agonist (acetylcholine; ACh)-evoked ($I_{evoked}$), and Gβγ-induced ($I_{βγ}$) GIRK currents. GIRK2 had a small $I_{basal}$[34], which was enhanced 4-8-fold by ACh (by activating the endogenous Gα$_{i/o}$βγ) and 30-60 fold by coexpressing nearly-saturating doses of Gβγ. In contrast, the non-prenylated Gβγ$_{C68S}$ neither activated GIRK nor affected $I_{evoked}$ (Fig. 1b–d, and Supplementary Fig. 1). We verified that N-terminally labeled YFP-Gγ and YFP-Gγ$_{C68S}$ were comparably expressed in whole oocytes and supported the expression of Gβ (Supplementary Fig. 1b, c). To assess PM localization, we immunostained Gβ in excised giant membrane patches (GMP)[32,43] using wild-type (WT) Gβ or an N-terminally myristoylated Gβ (myr-Gβ). Only WT Gγ (Gγ$_{WT}$), but not Gγ$_{C68S}$, supported GIRK2 activation and, correspondingly, PM enrichment of Gβ$_{WT}$ and myr-Gβ (Supplementary Fig. 1d–f).

These results confirm proper prenylation of Gγ in oocytes, which is essential for PM attachment of Gβγ and GIRK2 activation; but is it also involved in Gβγ interaction with GIRKs? We examined the interaction of purified, His-tagged Gβγ and Gβγ$_{C68S}$ with in vitro translated (ivt) Gβγ-binding proteins: Gα$_{i3}$; phosducin; cytosolic domains of GIRK1 and GIRK2 (G1NC and G2NC, respectively); and their truncated versions, G1NC$_{ΔdCT}$ and G2NC$_{trunc}$ (Fig. 1e). G1NC is a fusion protein of N- and C-terminal domains of GIRK1 (G1-NT and G1-CT). G1NC$_{ΔdCT}$ lacks the G1-dCT and binds Gβγ much weaker than G1NC[33] (Supplementary Fig. 2). G2NC$_{trunc}$ lacks the distal segments of the N- and C-terminal domains (G2-NT and G2-CT, respectively), as in structural and bilayer studies[23,44,45]. All ivt proteins bound Gβγ. Remarkably, lack of

prenylation dramatically reduced Gβγ interaction with Gα$_{i3}$ and phosducin, corroborating previous reports[36–38], and with all GIRK constructs (Fig. 1f, g), suggesting that Gγ prenylation directly contributes to Gβγ-GIRK interaction.

### Estimating Gβγ density in PM using calibrated fluorescence and quantitative Western blotting

We aspired to quantitatively analyze the membrane-delimited GIRK-Gβγ interaction in intact cells. To accurately calibrate protein surface density, we extended our previously developed calibration methods in Xenopus oocytes[35], which use two independent approaches.

The calibrated fluorescence (CF) approach measures the surface density of yellow, cyan or Split-Venus fluorescent proteins (YFP, CFP or SpV; collectively xFP), using xFP-labeled channels as molecular calipers. We used Gβγ-activated xFP-GIRK1/2[35], and additionally the constitutively active homotetrameric IRK1-YFP (Fig. 2a, b). Calibration involved expressing these channels at varying RNA doses, measuring whole-cell currents, and calculating the surface density of functional channels based on open probability ($P_o$), single-channel current ($i_{single}$) and cell's surface area[46] (Eq. 1 in Methods, Supplementary Fig. 3, Supplementary Table 3). YFP surface density was calculated assuming two or four YFP molecules per YFP-GIRK1/2 or IRK1-YFP channel, respectively. To avoid artifacts arising from any non-functional channels, we used channels' RNA doses in the 0.01–1 ng range, ensuring a linear relationship between fluorescence and whole-cell current and, accordingly, the calculated YFP surface density (Fig. 2a). Deviations were observed only at high levels of YFP-GIRK1/2 (5 ng RNA; Supplementary Fig. 6a). Additionally, comparing calibration with both YFP-GIRK1/2 and IRK1-YFP in the same experiment gave almost identical estimates of YFP surface density (Fig. 2a). Concomitantly, we expressed Gβ·$_{YFP}$Gγ (Gβ and YFP-Gγ) in separate groups of oocytes, measured YFP fluorescence at the oocyte's perimeter, and converted it to YFP-Gγ surface density with each caliper. The estimates of YFP-Gγ with both calipers showed strong linear correlation with a slope of 0.9 (Fig. 2b), validating the calibration protocol.

The CF procedure with Gβ·$_{YFP}$Gγ monitors YFP-Gγ rather than Gβ. We directly assessed the surface density of Gβ using the independent approach[35], quantitative Western blotting (qWB) of manually separated oocyte plasma membranes. We measured PM-associated Gβ with a Gβ antibody, using purified recombinant Gβγ for calibration (Fig. 2c, d). The PM density of the endogenous oocyte's Gβ was $30 \pm 13$ $\mu m^{-2}$, consistent with previous estimates[35] and comparable to ~$40$ $\mu m^{-2}$ in HEK cells[47]. Expressed Gβ surface levels were similar with either coexpressed Gγ or YFP-Gγ (Fig. 2e, and Supplementary Table 4). Overall, expressed surface Gβ (with 5 ng Gβ RNA) measured by qWB was $35 \pm 9$ $\mu m^{-2}$ ($n = 6$), about 2.5-fold lower than surface YFP-Gγ estimated by CF ($91 \pm 19$ $\mu m^{-2}$, $n = 7$, Fig. 2f). The difference is probably not related to methodology, because previously both CF and qWB gave similar estimates of 22-28 $\mu m^{-2}$ for a YFP-labeled Gβ[35]. Thus, evaluating YFP-Gγ may overestimate the coexpressed Gβ's surface density, possibly because YFP-Gγ associates with endogenous Gβ, or exists as a separate protein[48,49]. Therefore, we tested a variety of C- or N-terminally xFP-fused Gβ constructs (Supplementary Fig. 4). However, they yielded partial or no GIRK2 activation, and usually poorly activated GIRK1/2. SpV-Gβγ activated both GIRK1/2 and GIRK2 but induced smaller currents than Gβγ$_{WT}$. Only Gβ·$_{YFP}$Gγ activated GIRK channels like the Gβγ$_{WT}$[33].

We next varied expression levels of Gβ·$_{YFP}$Gγ and examined changes in surface densities of YFP-Gγ in intact oocytes and Gβ in GMPs (Fig. 2g). Reassuringly, surface levels of Gβ and YFP-Gγ were linearly correlated, with either GIRK2 or GIRK1/2 coexpressed (Fig. 2h, i). Thus, RNA dose-dependent changes in surface YFP-Gγ reflect corresponding changes in surface Gβ. Consequently, we routinely used Gβ·$_{YFP}$Gγ in the following experiments.

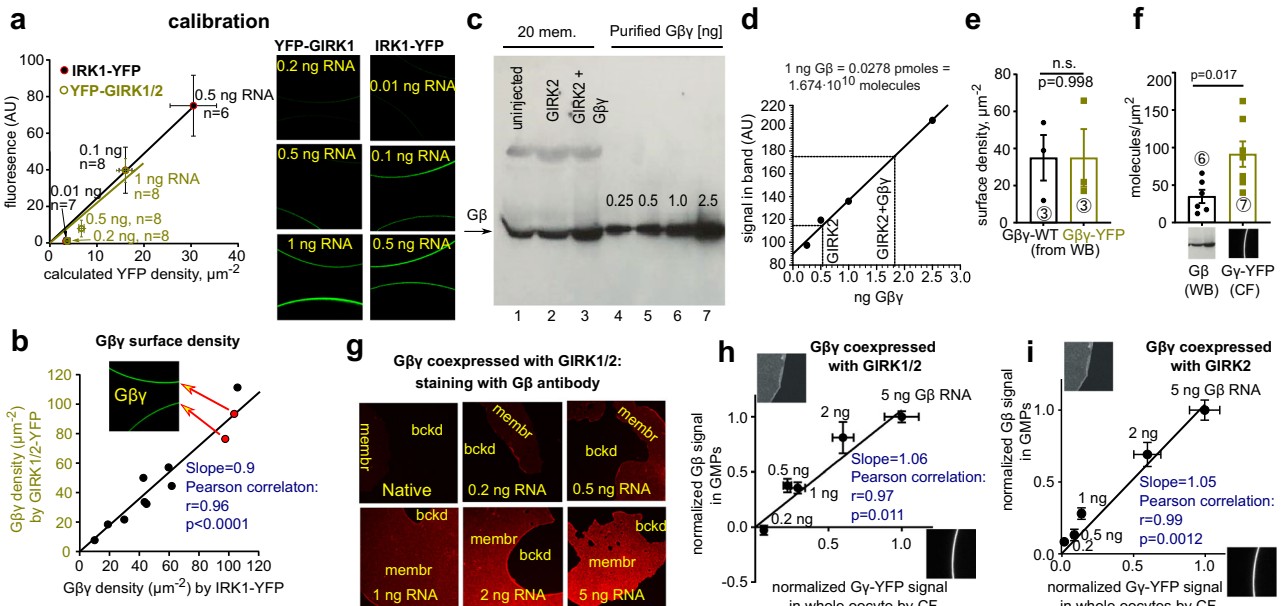

**Fig. 2 | Estimating Gβγ density in PM using calibrated fluorescence (CF) and quantitative Western blotting (qWB).** In oocyte experiments RNAs of YFP-Gγ and Gβ were injected at a constant ratio. **a** calibrating surface YFP-Gγ density with YFP-GIRK1/2 coexpressed with Gβγ (5:2 ng RNA/oocyte) or IRK1-YFP. Symbols show mean ± SEM. Number of oocytes (n) and amounts of channel RNA are shown near symbols. Surface density of channel-associated YFP was estimated from whole-cell currents. YFP fluorescence (in arbitrary units, AU) was measured from confocal images of intact oocytes (right panel). Image sizes are 272×272 μm. **b** calibration with either IRK1-YFP or YFP-GIRK1/2 gives similar estimates of surface density of Gβ·YFPGγ (same experiment in **a**). Data points represent individual oocytes. Inset shows representative oocytes (red symbols). Correlation was analyzed using two-tailed Pearson correlation and simple linear regression; p < 0.0001, r = 0.096. **c** measuring PM-attached Gβ (20 plasma membranes per lane) using WB with a Gβ antibody that well recognizes both endogenous and expressed Gβ[35], from naïve (uninjected) oocytes, or injected with GIRK2 RNA (2 ng) without or with Gβγ (5:2 ng RNA/oocyte). Lanes 4-7: calibration with recombinant Gβγ (0.25-2.5 ng/lane).

**d** estimating the amounts of Gβγ in PMs for lanes 1-3 from the calibration plot drawn using linear regression of data from lanes 4-7. **e** qWB-estimated surface density of Gβ, coexpressed with either Gγ or YFP-Gγ, is similar. Net amounts of Gβ were calculated in each experiment by subtracting the Gβ level of GIRK2-only expressing oocytes. 18–26 oocyte plasma membranes were loaded per lane. Bars show mean ± SEM. Statistics: two tailed unpaired t-test. Number of independent experiments is shown encircled in bars. **f** comparing the estimated levels surface density of YFP-Gγ (by the CF approach) and Gβ (by the qWB approach). Data with Gγ and YFP-Gγ were pooled. Statistics: unpaired t-test. **g** representative confocal images of GMPs (272 × 272 μm) from oocytes expressing Gβ, YFP-Gγ, and GIRK1/2 or GIRK2. Amounts of Gβ RNA are shown. **h, i** Gβ levels in GMPs and YFP-Gγ levels in intact oocytes are linearly correlated. Protein levels induced by different RNA doses were normalized to 5 ng Gβ in each experiment. Statistics: two-tailed Pearson correlation. Each point is mean ± SEM. Numbers of experiments and cells are shown in Supplementary Table 5.

## Affinity of Gβγ-GIRK2 interaction is in the low μM range

We investigated the dose-dependent activation of GIRK2 by Gβ·YFPGγ using the CF approach. We expressed GIRK2 with a range of Gβ·YFPGγ RNA doses. Following calibration (Fig. 3a), we quantified surface Gβ·YFPGγ density in individual oocytes and then measured single-channel $P_o$ in cell-attached patches of the same cells (Fig. 3b–d). The activation of GIRK2 was steeply Gβ·YFPGγ dose-dependent, with an initial slope of almost 3 on log-log coordinates (Fig. 3e). This indicates the requirement for ≥3 Gβγ molecules to open the channel, corroborating the WTM model[24] (Figs. 1a, 3e). Therefore, we analyzed the dose-response data using the WTM model version adjusted for real-cell conditions[15] (Fig. 3e, and Supplementary Fig. 8a #2, Methods Eq. 5) and, for comparison, the familiar but mechanistically less informative Hill equation (Eq. 4). We added to the equations a constant component (c) corresponding to $I_{basal}$. To convert the two-dimensional surface density to concentration, we used a standard procedure[24,35,50] assuming a submembrane 10 nm thick interaction volume.

Fitting the data with the WTM model (Fig. 3e) yielded cooperativity factor for each successive Gβγ binding (μ) of 0.44 and dissociation constant ($K_d$) of 44 Gβγ μm⁻² (7.4 μM). Fixing μ=0.3 as in Touhara et al.[15], yielded a $K_d$ of 17.3 μM, and Hill equation fit yielded a $K_d$ of ~4 μM (Fig. 3f). This is much lower than the 300 μM measured in bilayers even at saturating [Na⁺] of >20 mM[24], as highlighted with simulated dose-response curves in Fig. 3g.

Similar $K_d$ values were obtained for whole-cell currents of GIRK2 or HA-tagged GIRK2HA (which is activated by Gβγ like GIRK2[33,34],

Supplementary Fig. 1a). Fitting with WTM model (with fixed μ, reducing the number of free parameters) yielded $K_d$ of ~11 μM with μ=0.44 and ~31 μM with μ=0.3 (Fig. 4f; and Supplementary Table 6). GIRK2WT and truncated GIRK2 (as used in lipid bilayers) showed similar Gβγ sensitivity (one experiment; Supplementary Fig. 5).

## GIRK1/2 vs. GIRK2: higher apparent affinity to Gβγ and the role of Gβγ docking to GIRK1

Heterologously expressed GIRK1/2 has a high, Gβγ-dependent $I_{basal}$, contrasting the smaller, Gβγ-independent $I_{basal}$ of homotetrameric GIRK2[34,51,52]. Gβγ recruitment[33] and high $I_{basal}$ of GIRK1/2 and GIRK1/4 require an intact G1-dCT[34,52,53], suggesting that Gβγ docking at GIRK1 increases the local concentration of Gβγ around GIRK[4]. We hypothesized that this may also render higher apparent Gβγ affinity for GIRK1/2 compared to GIRK2.

We previously observed GIRK1/2 activation by expressing Gβγ at relatively low densities (5-50 μm⁻²)[35]. Here, we compared activation of GIRK2 and GIRK1/2 by Gβ·YFPGγ in the same experiment (Fig. 4a–c). Surface levels of YFP-Gγ and, subsequently, GIRK currents were measured in individual intact oocytes. Fitting these data with the WTM model revealed a significant difference between $K_d$ of GIRK2 and GIRK1/2 (45 and 9 μM, respectively, with μ=0.3, p = < 0.0001; Fig. 4b, and Supplementary Fig. 7a). Similar $K_d$ values were obtained for data grouped according to the RNA dosage (Fig. 4c). On average, the $K_d$ of GIRK1/2 was about 6-fold lower than GIRK2 (~5.5 μM vs. ~31 μM, p = 0.027, Fig. 4e, f, Supplementary Table 6). We also observed

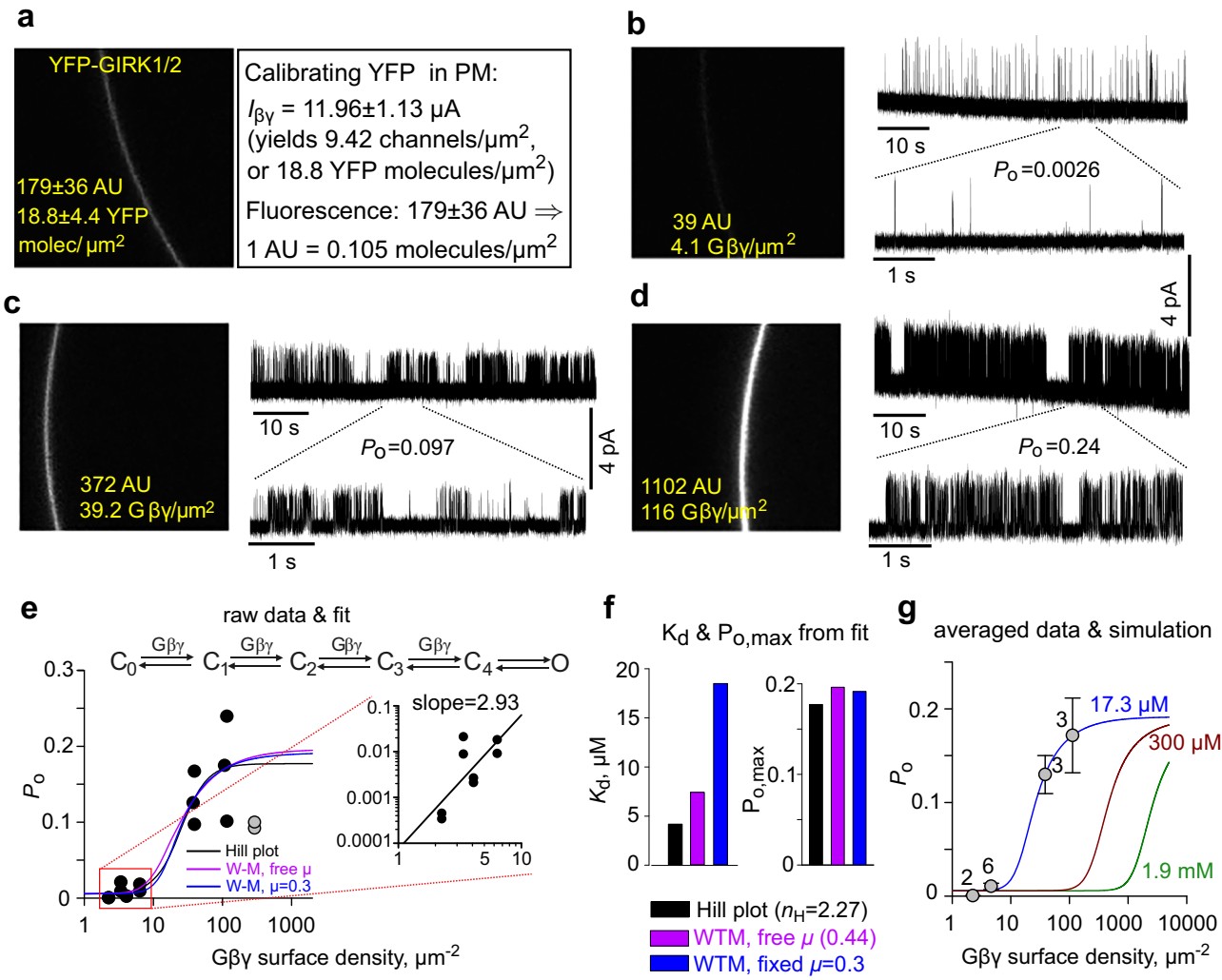

**Fig. 3 | Coexpressed Gβ·YFPGγ activates single GIRK2 channels with low-μM apparent affinity.** $P_o$ and Gβ·YFPGγ expression were measured in the same oocytes, injected with RNA of GIRK2 (25 or 50 pg/oocyte, ensuring low surface density), Gβ (0.2-20 ng/oocyte) and YFP-Gγ (40% of Gβ RNA). **a** calibration of surface density of YFP using YFP-GIRK1/GIRK2 (1 ng RNA each) coexpressed with WT-Gβγ (5:2 ng RNA, respectively). **b–d** representative confocal images of intact oocytes, and cell-attached patch records from these oocytes. **e**, changes in $P_o$ vs. estimated Gβ·YFPGγ PM density. Each circle represents $P_o$ measurement in a separate patch. Low $P_o$ observed in two patches from one oocyte (grey circles) with high surface Gβ·YFPGγ (290 μm⁻²) was attributed to Gβγ-induced desensitization, as reported previously for high [Gβγ] for GIRK1/4 and GIRK1/2[9,43]. These patches were excluded from fit. Lines show fits to Hill equation and to the WTM model, the latter with either fixed

(μ=0.3) or free cooperativity factor μ. *Inset* (right) shows the log($P_o$)-log[Gβ·YFPGγ] plot for the lowest Gβ·YFPGγ expression levels. The slope of the linear regression (black line) was 2.93. Hill coefficient ($n_H$) in the Hill plot fit was 2.37. The average Gβ·YFPGγ density at 5 ng Gβ RNA was 39.7 ± 6 μm⁻² (n = 12 oocytes). **f** $K_d$ and $P_{o,max}$ values from fits shown in e. For a full set of WTM fit parameters, see Supplementary Table 6. **g** simulated Gβγ dose-response curves with μ=0.3 and c=0.03, $P_{o,max}$ = 0.19, $K_d$ = 17.3 μM from the WTM fit of our data shown in (**f**) compared to values reported by Wang et al.[24]: $K_d$ = 1.9 mM for [Na⁺]in = 0 and $K_d$ = 300 μM for high [Na⁺]in (> 20 mM). For visualization purposes, $P_o$ values from patches with similar Gβ·YFPGγ levels were pulled and presented as mean ± SEM, with number of patches indicated next to each point.

an ~8-fold difference in $K_d$ of GIRK2-CFP and GIRK1/2-CFP (Supplementary Fig. 6). GIRK1/2 also exhibited the expected higher $I_{basal}$ than GIRK2. The basal current fraction (c) was higher in GIRK1/2 (~0.26) than in GIRK2 (0.02-0.03; Fig. 4e, f).

To investigate the role of Gβγ-anchor, we compared the Gβγ dose-dependence of GIRK1/2 to GIRK1$_{ΔdCT}$/2. GIRK1$_{ΔdCT}$/2 lacks the Gβγ-anchor, does not recruit Gβγ and has a reduced $I_{basal}$[33]. Remarkably, the $K_d$ of GIRK1$_{ΔdCT}$/2 was 9-fold higher compared to GIRK1/2 (Fig. 4d; p = 0.0003) and 3.8-fold higher in another experiment (Supplementary Fig. 7c; p = 0.0009). Thus, GIRK1's Gβγ-anchor contributes to the high apparent Gβγ affinity of GIRK1/2.

We added the c parameter to the original WTM model to account for $I_{basal}$. Instead of fitting c, $I_{basal}$ can be mechanistically explained and calculated using algorithms utilizing $I_{basal}$, $I_{evoked}$ and $I_{βγ}$ to estimate basal Gβγ and Gα in GIRK1/2 microenvironment[14,35]. We compared the

modified WTM (concerted cooperative), the graded contribution (channel opens with one Gβγ and sequential Gβγ binding progressively increases $P_o$[9,54]), and two non-cooperative models (Fig. 4g, and Supplementary Methods, Supplementary Fig. 8). With each model, we calculated basal Gα, Gβγ and $I_{basal}$ for a range of $K_d$ values, and subsequently simulated dose-response curves for expressed Gβγ with μ=0.3. Both cooperative models matched the experimental data with $K_d$ between 1-10 μM (Fig. 4g). Expectedly, the non-cooperative models predicted lower $K_d$. The cooperative models also provided stable estimates of basal Gα and Gβγ across a wide $K_d$ range, 0.1-30 μM (Supplementary Fig. 8).

The interactions of Gβγ with the PM and the channel are reversible. Therefore, we expected that removing the cytosolic Gβγ reserve by excising a membrane patch into a Gβγ-free solution would reduce the PM- and GIRK-associated Gβγ, deactivating GIRK channels. We

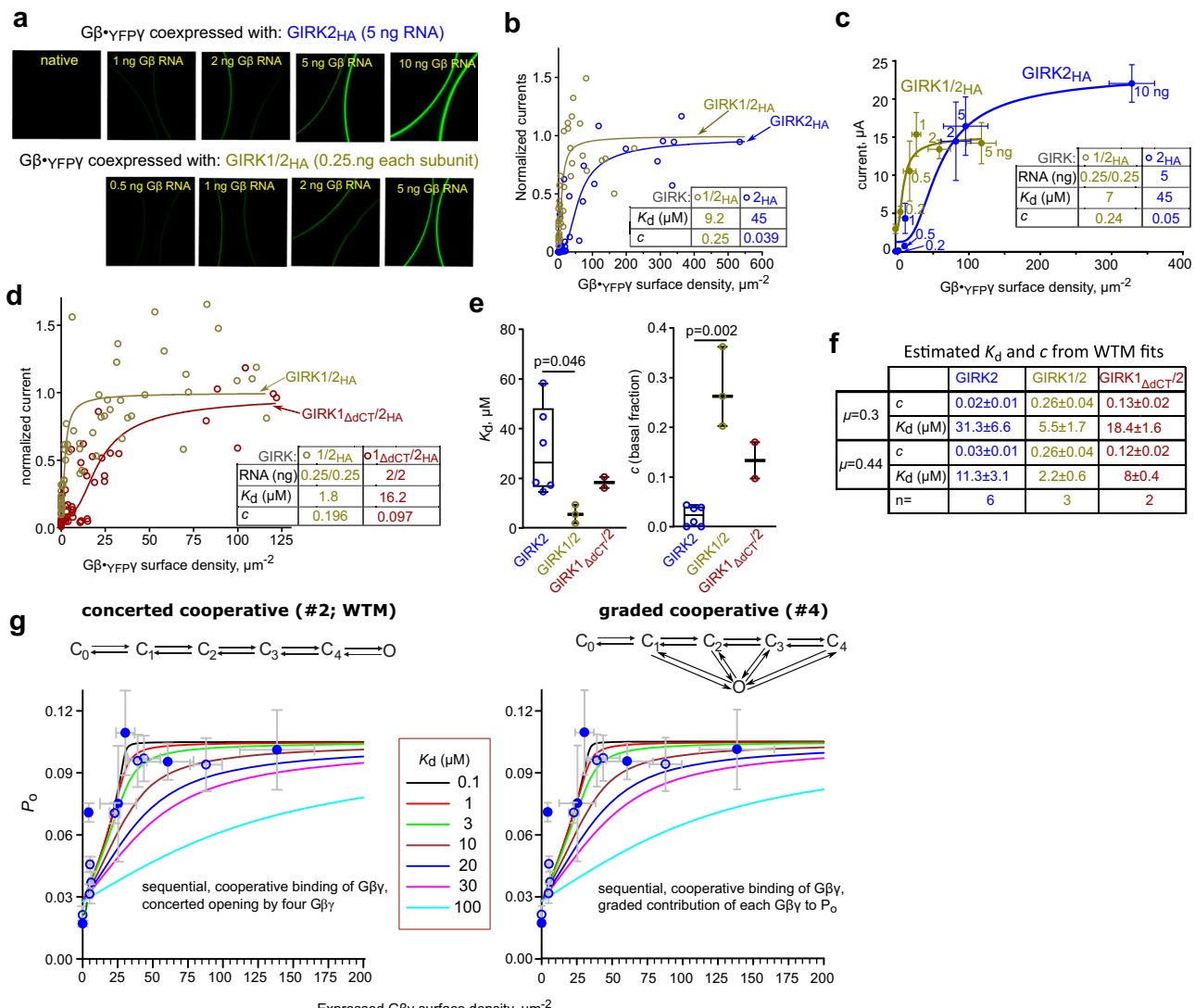

**Fig. 4 | GIRK2 and dCT-truncated GIRK1 show lower apparent affinity to Gβγ than GIRK1/2. a–d** GIRK2_HA was used in these experiments. Gβ:YFP-Gγ RNA ratio was 2:1. RNA doses of GIRKs and WTM fit parameters are shown in insets in b-d. Surface density of YFP was calibrated using IRK1-YFP. Currents were measured in 24 mM [K⁺]_out. **a–c** dose-dependent activation of GIRK2_HA homotetramers and GIRK1/2_HA heterotetramers by Gβ·_YFP_Gγ (experiment #4). **a** examples of confocal images (272×272 μm) in oocytes expressing Gβ·_YFP_Gγ with GIRK1/2_HA or GIRK2_HA. **b** dose-dependent activation of GIRK1/2_HA and GIRK2_HA by Gβ·_YFP_Gγ. Each point represents an individual oocyte. Currents were normalized to the maximal $I_{βγ}$ ($I_{max}$, Supplementary Table 6) and fitted to the WTM model (with μ = 0.3). The differences between the fitted $K_d$ were significant (F(1, 81) = 18.95, p < 0.0001). See additional analysis in Supplementary Fig. 7a. **c** results of the same experiment were analyzed for groups of oocytes according to the amount of Gβ RNA (shown near each point).

Data are presented as mean ± SEM of $I_{βγ}$ and YFP-G_γ; numbers of oocytes are shown in Supplementary Table 5. **d** dose-dependent activation of GIRK1/2_HA and GIRK1_ΔdCT/2_HA by Gβ·_YFP_Gγ. (Experiment #7; additional details in Supplementary Fig. 7b). Analysis and presentation of data are as in b. The differences between fitted $K_d$ were significant: F(1, 103) = 14.18, P = 0.0003. **e, f** summary of parameters of the WTM fit with fixed μ=0.3 for all experiments (**e**) and with μ=0.3 or μ=0.44, presented as mean ± SEM (**f**). Statistics in (**e**) unpaired two-tailed t-test between GIRK2 and GIRK1/2. Box shows 25th–75th percentiles; whiskers, min–max; line, median. See Supplementary Table 6 for full details. **g** simulation of GIRK1/2_HA activation by Gβγ with a range of $K_d$ values (solid lines) with the cooperative models (Supplementary Fig. 8a). The simulated curves are superimposed on data, shown as mean ± SEM, from experiments #4 (closed circles) and #7 (open circles). Full details, including n, are in Supplementary Fig. 8c.

anticipated slower deactivation in channels with a high-affinity Gβγ-anchor.

To test this hypothesis, we recorded Gβγ-activated channels in cell-attached patches and then excised them into an ATP and Na⁺-containing bath solution (Fig. 5a–d). GIRK1/2 activity decayed (deactivated) slowly, with 30-50% persisting after 5 minutes (Fig. 5a, d). The decay followed a single exponent with a time constant (τ) of >2 min and a non-deactivating fraction (C) of 0.34. In contrast, GIRK2 and GIRK1_ΔdCT/2 exhibited faster and more complete decay (Fig. 5b–d, f). Excising patches into an ATP-free solution, which could deplete PIP₂ in the PM[55], had a minimal impact on GIRK2 and GIRK1_ΔdCT/2 decay, and

slightly affected GIRK1/2 (Fig. 5e, f). This suggests that GIRK deactivation is mainly governed by the depletion of Gβγ associated with or surrounding the channel, rather than PIP₂ depletion.

**G1-NT and G1-dCT form a Gβγ-binding site and contribute to channel's interaction with Gγ's prenylation tail, Gγ_prenyl**

Although deleting G1-dCT thwarts Gβγ binding, G1-dCT alone does not strongly bind Gβγ[56], indicating that the Gβγ-anchor includes additional Gβγ-binding segments[33]. To identify them, we scanned arrays of overlapping peptides covering the cytosolic domains of GIRK1 and GIRK2 (G1NC, G2NC) for His-Gβγ binding (Figs. 1e, 6a–c, and

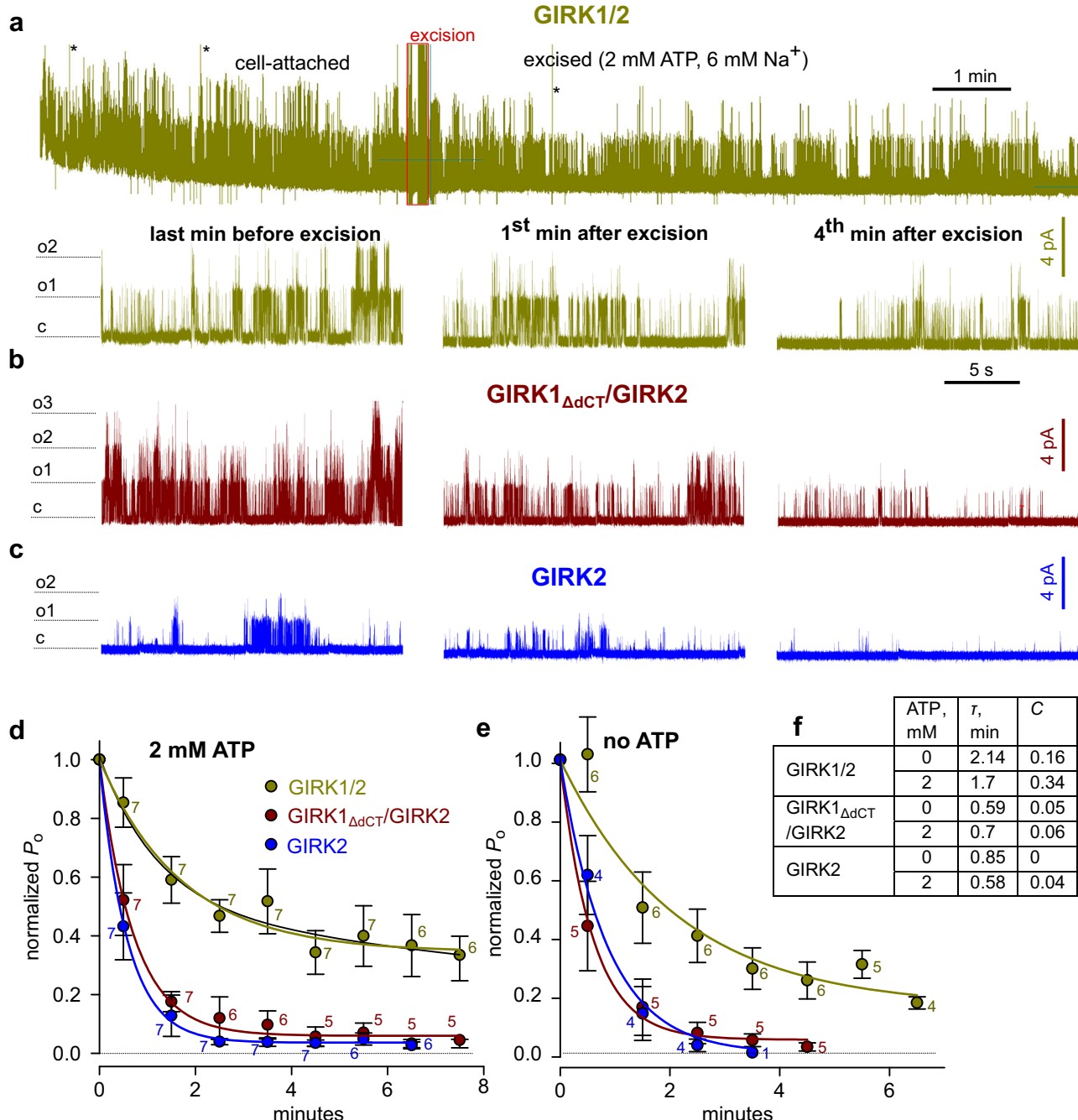

**Fig. 5 | Different patterns of deactivation of GIRK2 and GIRK1/2 after patch excision and the role of G1-dCT.** Channels were expressed at low densities, with a high dose of Gβγ or SpV-Gβγ (5 ng Gβ and 1 ng Gγ). **a** representative recording of GIRK1/2. *Top*, the complete original recording that lasted 13.5 min. After ~4 min in cell-attached mode, the patch was excised into bath solution containing 2 mM ATP and 6 mM NaCl, causing a gradual decay of activity. *Bottom*, zoom on 20 s segments of the record during the indicated times before and after excision. **b**, **c** similar stretches from recordings of representative GIRK1_{ΔdCT}/2 and GIRK2 recordings. **d** time course of deactivation after excision summarized as $NP_o$ within consecutive 60 s segments of record, normalized to $NP_o$ during the last minute before excision. ($NP_o$ is a measure of total activity in the patch, i.e., number of channels times $P_o$). Each point is mean ± SEM, with number of patches shown near each symbol. Lines show single-exponential fits; fitting with two exponents did not produce better results (exemplified for GIRK1/2 with ATP, black line). **e** similar results were obtained when the patches were excised into an ATP-free solution. Data presentation as in (**d**). **f** comparison of exponential fit parameters for the three channel types, with and without ATP. τ is the time constant of the exponential decay and C is the extrapolated non-deactivating fraction.

Supplementary Fig. 9). Scanning revealed three Gβγ-binding segments mainly overlapping the C1 and C3 segments from previous biochemical studies[28,56]. Two segments fully (in GIRK2) or partially (in GIRK1) overlapped the Gβγ-binding amino acid (a.a.) clusters from the crystallized GIRK2/Gβγ complex[5] (Fig. 6d). Additionally, Gβγ bound to segments in G1-NT (a.a. ~20-50), parts of G1-dCT (a.a. ~390-440 and ~485-501), and G2-NT and G2-dCT.

If a separate GIRK1's Gβγ-binding segment combines with G1-dCT to form the Gβγ-anchor, deleting it from G1NC should reduce Gβγ binding. We used prenylated His-Gβγ to pull-down the full-length *ivt* G1NC or G1NC with specific segment deletions, and a fusion protein of G1-NT and G1-dCT, G1NdCT (Fig. 7). Gβγ binding was unaffected by the deletion of internal segments C1-C3 and tended to decrease after the deletion of G1-NT (G1CT construct). G1-dCT and G1-NT showed weak

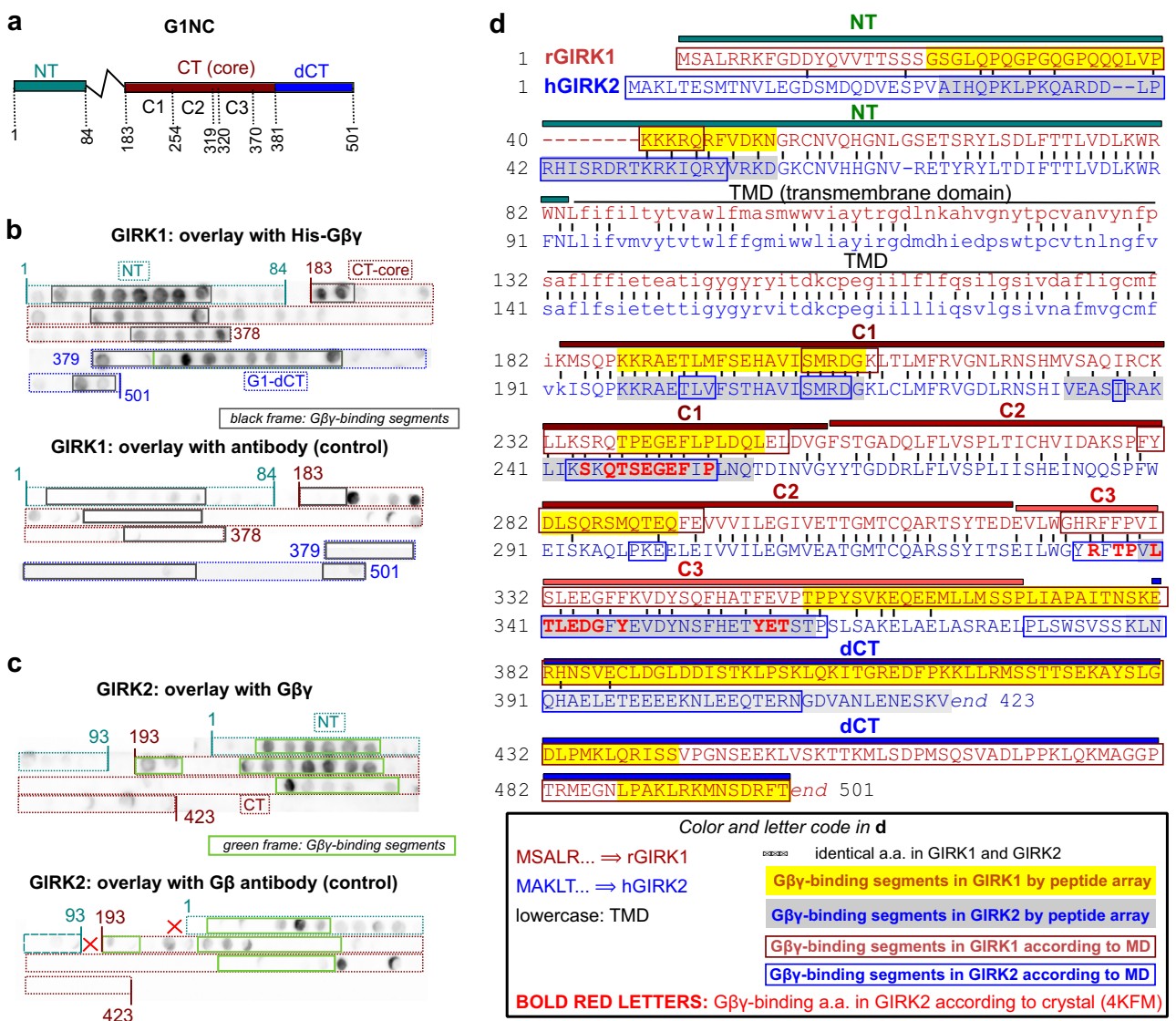

**Fig. 6 | Peptide array scanning for Gβγ binding sites in the cytosolic domains of GIRKs. a** linear scheme of G1NC incorporating segment names (NT, CT, etc.) and a.a. numbers illustrating the design of the peptide array (b) and the constructs used in pull down experiments of Fig. 7. **b, c** arrays of 25-mer overlapping peptides with a 5 a.a. shift of G1NC (b) and G2NC (c), spotted onto a membrane. Upper images show overlays with purified His-Gβγ, probed with the Gβ antibody (4 experiments for G1NC, 3 for G2NC). Gβγ-binding segments are enclosed within solid-border rectangles. Bottom images show control arrays overlayed with Gβ antibody only (two experiments for each channel). In GIRK2 some non-specific labeling (without Gβγ) was observed in segments designated as Gβγ-binding. The non-specific labeling was weaker and appeared in fewer spots, therefore we have not discarded these spots from the area assigned as Gβγ-binding. **d** alignment of rGIRK1 (rat GIRK1) and hGIRK2 (human GIRK2) a.a. sequences used in peptide array scans. The Gβγ-binding segments suggested by peptide arrays are highlighted in yellow (GIRK1) and gray (GIRK2). A weakly labeled potential Gβγ-binding segment in the distal CT of hGIRK2 is labeled with a lighter gray background. Gβγ-binding segments suggested by molecular dynamics (MD) simulations (from Fig. 8) are framed by dark red (GIRK1) and blue (GIRK2) rectangles. Amino acids in GIRK2 that make contacts with Gβγ according to the crystal structure of the GIRK2-Gβγ complex, 4KFM[5], were determined using the Prodigy software (https://rascar.science.uu.nl/prodigy/) and are highlighted in bold red letters.

and negligible Gβγ binding, respectively. However, both G1NdCT and the fusion of N-terminal a.a. 40-84 with G1-dCT, G1N(40-84)dCT, strongly bound Gβγ, suggesting that the GIRK1's Gβγ-anchor comprises G1-dCT and part(s) of G1-NT.

We conducted coarse-grain (CG) and all-atom molecular dynamics (MD) simulations to further investigate the involvement of G1-NT, G1-dCT and Gγ_prenyl in GIRK-Gβγ interactions. These elements are missing from the available high-resolution structures. We modeled full-length and truncated G1NC and G2NC tetramers complexed with Gβγ using AlphaFold3 and manually added the prenylation tails (Fig. 8, and Supplementary Fig. 10a, Supplementary Table 11). The initial CG system included four Gβγ bound to a G1NC or G2NC tetramer without the PM and bulk Gβγ in the cytosol. MD simulations accurately

captured the two Gβγ-interacting surfaces from the GIRK2-Gβγ crystal structure[5] and predicted additional Gβγ-binding segments, most of which showed excellent (in G2NC) or considerable (in G1NC) agreement with peptide arrays (Figs. 6d, 8b, and Supplementary Fig. 10b), lending credibility to the combined analysis. Further analysis revealed that Gγ_prenyl spent 100% of the simulation time interacting with G1NC, mainly with the beginning of G1-NT, as compared to only 6.4% ± 1.3 with G2NC (p = 0.039; Fig. 8c, d; Supplementary Tables 12, 14). Gγ_prenyl likely accounts for most of the Gβγ binding to the first G1-NT segment predicted by the MD (compare Fig. 8b, c), explaining the poor Gβγ labeling of a.a. 1-25 in peptide array overlays, where solid support-spotted peptides may be less accessible to Gγ lipid moiety. Additionally, Gγ_prenyl also interacted with hydrophobic a.a. in the C-terminus of

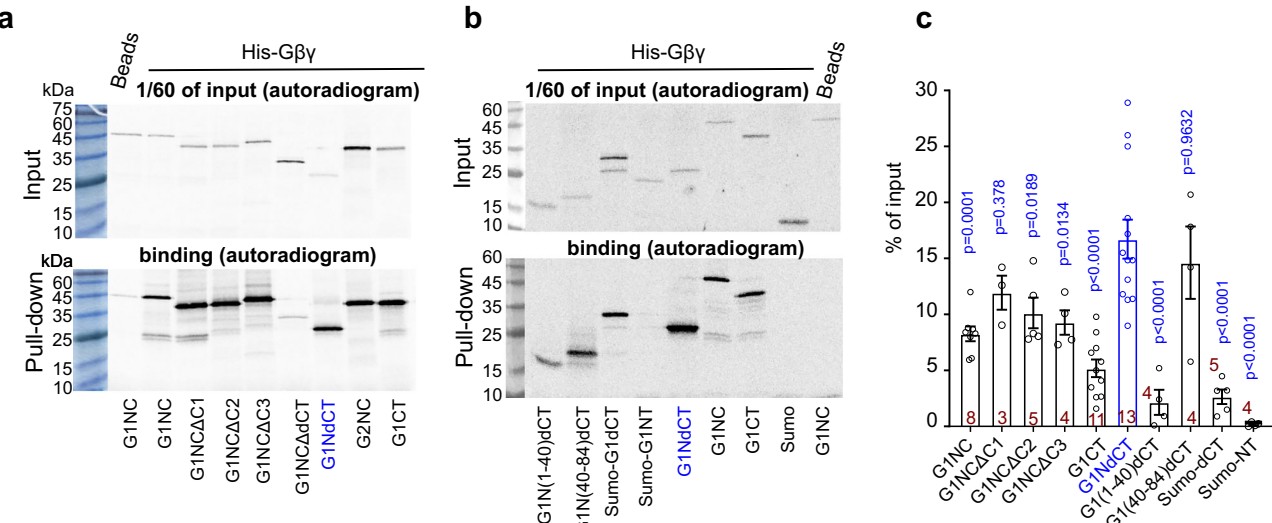

**Fig. 7 | Fused G1-NT and G1-dCT of GIRK1 form a high-affinity Gβγ-binding site.**
**a, b** SDS-PAGE autoradiograms of pull-down of [35Met]-labeled *iut* G1NC, G1NC-derived constructs and additional controls by His-GβγWT from two representative experiments. G1-NT and G1-dCT were fused to Sumo for stability. **c** summary of pull-down experiments. Binding of each construct was calculated as percentage of input of that construct in the same experiment. Each bar represents mean ± SEM; number of independent experiments are shown within the bars. Statistics: One Way ANOVA followed by Dunnet's multiple comparison method vs. control group, G1NdCT. *p* values are shown above the bars. Statistics for G1NC comparisons are presented in Supplementary Table 7.

Gβ (Supplementary Table 13). Backmapped atomistic simulations of G1NC yielded results consistent with the CG simulations, further supporting the robustness of the approach (Supplementary Fig. 10c). In simulations where PM was included, Gγprenyl interactions remained stable, supporting the robustness of the binding site predictions (Fig. 8d). Our simulations started with Gβγ pre-bound to the channel; fully ab initio simulations would require significantly longer sampling but could potentially reveal additional membrane interactions of Gγprenyl. However, such analyses are beyond the scope of the current study.

Deleting G1-dCT abolished Gγprenyl interaction with the remainder of G1NC, G1NCΔdCT (*p* = 0.0003), whereas truncated G2NC retained Gγprenyl interaction (Fig. 8c, and Supplementary Table 12). Remarkably, Gγprenyl interaction with the first segment of G1-NT was lost upon G1-dCT deletion (Fig. 8c), reinforcing the idea that G1-NT and G1-dCT form a Gβγ-binding unit. MD simulations also revealed details of the GIRK1's NT-dCT structural unit, with segments of a.a. 27-31 (NT) and ~450-460 (dCT) interacting 99% of the simulation time (Fig. 8e, f, Supplementary Data 1). Notably, this NT-dCT unit is not predicted in the G1NC-Gβγ model by AlphaFold but assembles dynamically during the simulation. In support of the important role of the NT-dCT unit for GIRK1 interaction with Gγprenyl, we observed a complete loss of G1NdCT-Gβγ binding with the non-prenylated GγC68S (Supplementary Fig. 11).

## Discussion

In this study we address two key issues in the GPCR-Gαβγ-GIRK signaling cascade: the Gβγ-GIRK interaction affinity and the subunit-dependent GIRK-Gβγ preassociation. We hypothesized that Gγ prenylation contributes to Gβγ-GIRK interactions and demonstrated that elimination of prenylation thwarts Gβγ interaction with cytosolic domains of GIRK1 and GIRK2 (G1NC and G2NC; Fig. 1). Expectedly, PM targeting was also abolished (Supplementary Fig. 1). However, membrane targeting was not involved in our Gβγ binding assays, performed in membrane-free detergent solutions. The importance of Gγprenyl in full channel context in PM is supported by higher GIRK2-Gβγ affinity in intact oocytes (Figs. 3, 4) compared to non-prenylated Gβγ in bilayers[24]. We conclude that, besides its well-established role in membrane attachment of Gβγ, Gγ prenylation enhances Gβγ-GIRK

interaction, as in many other Gβγ interactors[36–42]. The mechanism could involve transient interactions of Gγprenyl with hydrophobic sites in Gβγ's partner[57] or Gβ itself, stabilizing the conformation favoring Gβγ function[38,40,42,58]. In support, MD simulations reveal interactions of Gγprenyl with both, specific sites in GIRK1 and GIRK2, and C-terminal hydrophobic residues of Gβ (Fig. 8, and Supplementary Fig. 10, Supplementary Tables 12-14).

The dual role of Gγ prenylation complicates the interpretation of in vitro affinity measurements. Measuring GIRK's $K_d$ in excised PM patches with prenylated Gβγ in bath solution grossly overestimates affinity ($K_d$ = 2-11 nM, Supplementary Table 8) due to Gβγ's preferential partitioning to the PM. We addressed the challenge of quantitating GIRK activation by Gβγ in intact cells utilizing *Xenopus* oocytes, which are highly suitable for accurate titration and monitoring of expression and function of membrane proteins. We constructed Gβγ-GIRK dose-response relationships by varying Gβγ expression and measuring surface densities of Gβγ and GIRK responses. Our results support the WTM model[15,24] of collision-coupled, cooperative activation of GIRK2 by four Gβγ molecules. However, our affinity estimates are substantially higher.

$K_d$ estimates rely on accurate calibrations used to measure Gβγ surface levels. We validated our CF calibrations using two molecular calipers, YFP-GIRK1/2 and IRK1-YFP (Fig. 2). These results, along with previous compatibility tests between CF and qWB methods[35], enhance confidence in both calibration procedures. Importantly, only prenylated Gβγ dimer reaches the PM and is captured in our measurements of surface Gβγ, irrespective of total prenylated/non-prenylated cellular Gβγ content. The CF approach is advantageous for measuring expression and function of fluorescently labeled proteins in individual, intact cells. Disappointingly, xFP-Gβ constructs poorly activated GIRKs, especially GIRK2, calling for caution in using xFP-labeled Gβ in functional studies. Consequently, in most dose-response experiments we used Gβ·YFPGγ, which activated GIRKs like GβγWT. When expressing Gβ·YFPGγ, the surface densities of Gβ and YFP-Gγ were linearly related, but measuring YFP-Gγ might overestimate coexpressed Gβ, and accordingly the $K_d$, by up to 2.5-fold (Fig. 2). To avoid over-interpretation, we did not apply the YFP-Gγ correction (for measuring YFP-Gγ as a proxy for Gβγ) in our tables and figures.

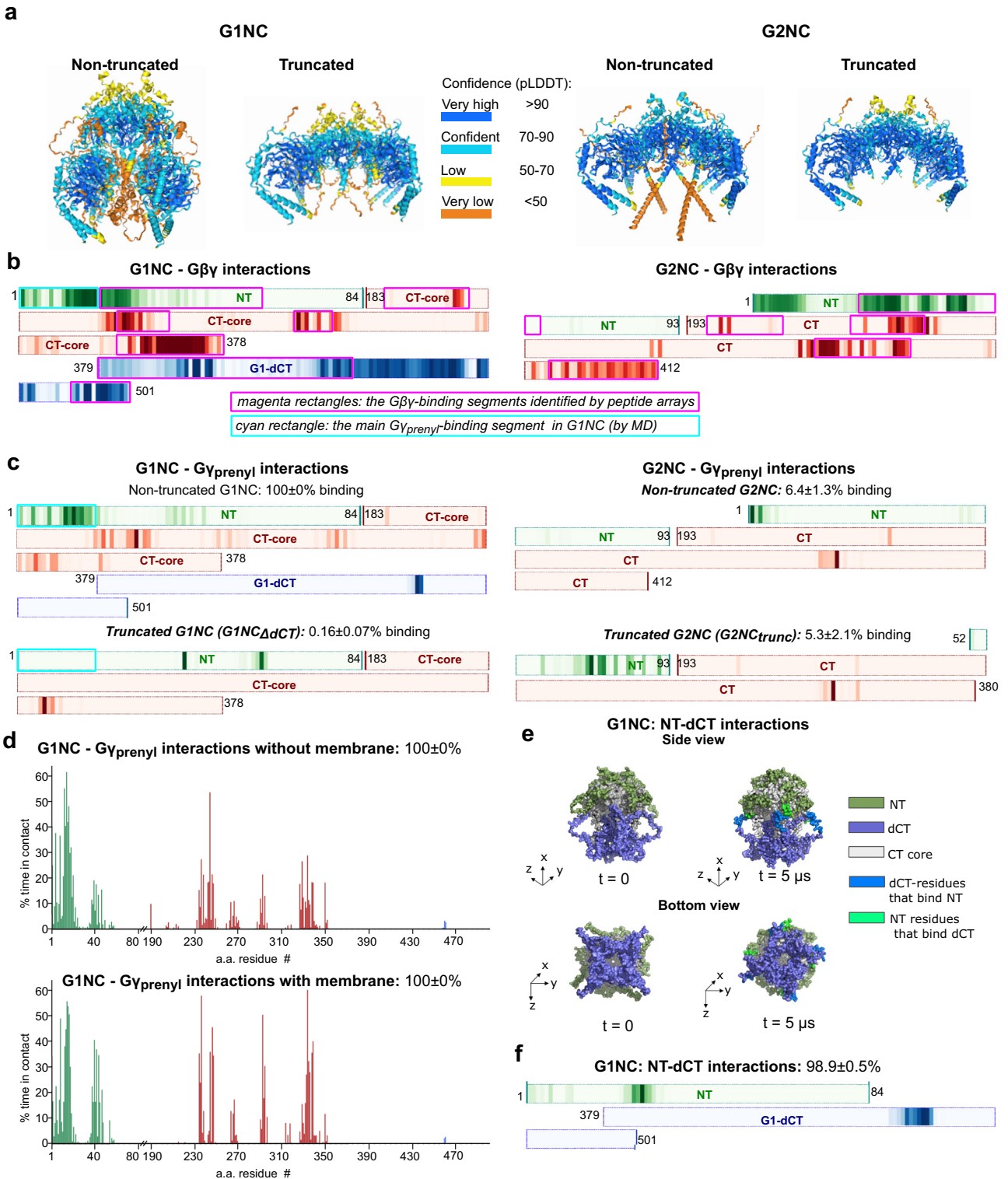

Even before formal curve fitting, the Gβγ-GIRK2 dose-responses clearly show that only 10 to 150 μm⁻² of free Gβγ is needed for 10% to 80-90% GIRK2 activation in intact oocytes (Figs. 3, 4), much less than the >1200 μm⁻² predicted by bilayer results[24]. Applying the 2.5-fold YFP-Gγ correction shifts the activation range to 4−60 μm⁻². We propose that the higher affinity that we find is mainly due to Gγ prenylation. Interestingly, GIRK2's $I_{evoked}$ (via m2R) is only 10% of Gβγ-evoked (Supplementary Fig. 1a). Thus, activation of endogenous $G_{i/o}$ ($Gα_{i/o}βγ$) releases 10-15 molecules/μm⁻² of free Gβγ, corresponding to 30−50% of total endogenous Gβγ in oocyte's PM, ~30 μm⁻² (Fig. 2). Importantly,

coexpressing $Gα_{i3}$ and Gβγ with m2R yields $I_{evoked}$ matching $I_{βγ}$[59]. Clearly, endogenous $G_{i/o}$ is insufficient to activate all GIRKs; but m2R can activate all channels when enough $Gα_{i/o}βγ$ is present.

Comparing $K_d$ for a multistep cooperative reaction is complex, even with the same kinetic model. The $K_d$ derived from dose-response data is interdependent with the Gβγ cooperativity factor μ: higher μ gives a lower $K_d$. μ is Na⁺-dependent[24] but can be considered constant at stable cytosolic [Na⁺][15]. (We consider [Na⁺]$_{in}$ in oocytes, 10-20 mM, as close to saturating for GIRK2).

**Fig. 8 | MD simulations corroborate the role of G1-NT and G1-dCT in interactions with Gβγ and the prenylation tail, Gγ$_{prenyl}$. a** the initial AlphaFold 3 models of complexes of G1NC and G2NC with prenylated Gβγ (see Supplementary Table 11 for further details). **b** heatmaps illustrating the G1NC and G2NC residues contributing to Gβγ binding. CG analysis was carried out on five 5-μs production runs for G1NC and ten for G2NC. Darker coloring corresponds to greater overall contacts between the channel and Gβγ across all production runs. The magenta rectangles superimposed onto the heatmaps correspond to the Gβγ-binding segments identified by the peptide arrays (Fig. 6). The cyan rectangle outlines the main Gγ$_{prenyl}$-binding segment, the beginning of G1-NT. **c** heatmaps of interactions of G1NC and G2NC and their truncated versions with Gγ$_{prenyl}$. % binding is the percentage of time when at least one prenylation tail is bound to the channel. Note that the Gγ$_{prenyl}$

interaction with the most prominent site, a.a. 1-20 of G1-NT (cyan rectangle), is lost after G1-dCT removal (details in Supplementary Table 13). **d** the histograms show % of time spent by G1NC a.a. residues in contact with the Gγ$_{prenyl}$ in simulations without membrane (top; 5×5-μs runs) and with added POPC (1-Palmitoyl-2-oleoyl-sn-glycero-3-phosphocholine) membrane (bottom; 3 × 5-μs runs). **e, f** the interaction between G1-NT and G1-dCT in G1NC. A frame with a contact was defined as one in which at least one G1-dCT chain is bound to the G1-NT, with a cutoff of 6 Å. G1-NT and G1-dCT were in contact in 98.9 ± 0.5% of the frames in the five runs. The structures of G1NC (**e**) are shown at the beginning and at the end (1 μs) of a representative run. Areas of contact are highlighted. The heatmap (**f**) indicates that the main interaction segment in G1-NT is a.a. 25-32. Full details of all analyses are provided in Supplementary Tables 11–14.

Our average $K_d$ estimates for GIRK2 are 11 μM with μ=0.44 (from Fig. 3) and 31 μM with μ=0.3[15]. These are likely overestimates, for two reasons. First, Hill and WTM models assume ligand excess over receptors. This is uncommon in cellular protein-protein interactions, leading to ligand depletion and $K_d$ overestimation: more receptors (GIRK) mean less free ligand (Gβγ) per receptor[60]. This is relevant to our whole-cell experiments, where GIRK2 surface density was 17 ± 5 μm$^{-2}$ (Supplementary Table 6), comparable to the functional Gβγ range. Second, applying the 2.5-fold YFP-Gγ correction would shift $K_d$ to 4-12 μM, quite close to the most accurate in vitro measurement available for prenylated Gβγ, 0.8 μM (interaction with CT of GIRK4, by surface plasmon resonance)[21].

Notably, less Gβγ is needed for GIRK1/2; 50 μm$^{-2}$ yields full activation (Fig. 4), confirming previous results[35]. The 10-15 Gβγ molecules/μm$^2$ released by GPCR activation would yield $I_{evoked}$ of about 50% of $I_{βγ}$ (Fig. 4), consistent with experiments[35]. Accordingly, GIRK1/2's apparent $K_d$ from WTM fits is 5-6-fold lower than GIRK2's. We further analyzed the GIRK1/2 dose-response data by including explicit calculations of Gα and Gβγ needed to produce the observed $I_{basal}$ and $I_{evoked}$[14,35]. Across a broad $K_d$ range (0.1 to 10 μM), the two cooperative models (Fig. 4g) predicted that both $I_{basal}$ and $I_{evoked}$ could be generated by physiologically relevant amounts of 1-2 Gα and 3-4 Gβγ per channel (Supplementary Fig. 8). This corresponds to less than 40 μm$^{-2}$ of Gβγ assuming physiological densities of GIRKs (2-10 μm$^{-2}$)[16,26].

GIRK1's Gβγ docking site (anchor) emerges as the major factor determining the higher affinity of GIRK1/2. This is suggested by (i) the 4-9-fold affinity drop in GIRK1$_{ΔdCT}$/2, which lacks the main part of the anchor, G1-dCT[33] (Fig. 4); (ii) the fast deactivation after patch excision of GIRK1$_{ΔdCT}$/2, mirroring GIRK2, indicating faster Gβγ dissociation (Fig. 5). These results, along with the preservation in GIRK1$_{ΔdCT}$ of Asn-217 that renders GIRK1 Na$^+$-insensitive[61], imply a minor role for the differences in Na$^+$-dependence of Gβγ affinity in GIRK1 and GIRK2[25] in our experiments. The anchor probably increases the apparent affinity through local enrichment of Gβγ (see below).

We proposed that Gβγ-anchors are distinct from the Gβγ-binding "activation" sites, which induce channel opening[4] and are located at the interface between core-CTs of two adjacent GIRK subunits[5,62]. Removal of G1-dCT preserves maximal Gβγ activation and $P_o$ but eliminates Gβγ recruitment and high $I_{basal}$[33], suggesting functional separation of docking and activation. Structural separation is suggested by strong Gβγ binding to G1NC that persists after removing major components of the activation site (C1-C3) and even the whole core-CT, leaving only the fused NT and dCT (G1NdCT) (Fig. 7). Thus, the anchor dominates the overall Gβγ affinity of GIRK1's cytosolic domain and does not include elements from core-CT. Both G1-NT and G1-dCT bind Gβγ[56,63] (Fig. 6) but much weaker than their fusion protein, G1NdCT (Fig. 7). These results suggest that the Gβγ-anchor is formed jointly by G1-NT and G1-dCT. Interestingly, truncation of G2NC did not significantly reduce Gβγ binding, and the functional impact was minor (Fig. 8c, and Supplementary Figs. 2, 5). However, adding G1-dCT to GIRK2 increased $I_{basal}$ and conferred Gβγ recruitment[33], suggesting that G2-NT may form Gβγ anchors with G1-dCT.

Peptide array scan and MD simulations provide additional insights. Both approaches identify known Gβγ-binding sites in core-CT, and NT and dCT Gβγ-binding sites in GIRK1 and GIRK2. Our MD analysis used AlphaFold-models including unstructured but essential elements absent from crystal structures: Gγ$_{prenyl}$ and GIRKs' NT and dCT. Despite the low-confidence of AlphaFold predictions for some of these elements, MD calculates interactions based on physical parameters and can capture dynamic interactions even if the initial structure is uncertain. Importantly, congruent with experimental results (Fig. 7, and Supplementary Fig. 11), the simulations reveal a dynamically arising structural unit formed by G1-NT and G1-dCT, and extensive interactions of Gγ$_{prenyl}$ with G1NC, particularly G1-NT, and some with G2NC (Fig. 8). Remarkably, Gγ$_{prenyl}$–G1-NT interaction is lost, and Gβγ–G1-NT interaction is reduced after deleting G1-dCT, although G1-dCT itself barely interacts with Gγ$_{prenyl}$. These results corroborate the idea that the Gβγ-anchor is a standalone structural and functional unit formed by G1-NT and G1-dCT, with G1-dCT essential for its integrity. Notably, Gγ assists Gβ in GIRK activation[49,58]. Gγ$_{prenyl}$-anchor interaction may also be involved, since removing G1-dCT or Gγ's C-terminal region, which includes the prenylation site, eliminates Gγ's enhancing effect[49,58].

Figure 9 summarizes our view of Gβγ-GIRK2 vs. Gβγ-GIRK1/2 interactions, gating, and the anchor's role. The dynamic equilibrium between channel-bound, membrane-associated and cytosolic Gβγ determines the local Gβγ concentration in channel's microdomain. Free Gβγ can reversibly partition from the cytosolic reserve to the PM, activating GIRKs. Comparing $K_d$ for GIRK1/2 activation by Gβγ in whole oocytes (Fig. 4f) and excised patches[43] yields a Gβγ PM/cytosol partition coefficient between 140 and 425 (Supplementary Fig. 12), close to earlier estimates of ~300[64].

Our findings confirm that GIRK2 is gated through collision-coupling with Gβγ, cooperative Gβγ binding, and concerted activation by four Gβγ occupying all activation sites (Fig. 9a), consistent with the WTM model[15,16,24]. However, in intact *Xenopus* oocytes (with [Na$^+_{in}$] between 10-20 mM), the Gβγ-GIRK2 affinity is significantly higher than the bilayer estimates, primarily due to Gγ prenylation, which enhances Gβγ functionality and interaction with GIRKs. The high affinity guarantees efficient G$_{i/o}$-GIRK2 signaling without the need for obligatory hotspots to account for physiological response (although we cannot exclude hotspots or crowding in oocyte PM, which could upshift our $K_d$ estimates).

In distinction, GIRK1/2 operates within a more complex dynamic system featuring two kinds of binding sites, docking (Gβγ-anchors) and activating. The anchor, formed jointly by G1-NT and G1-dCT, is functionally and topologically separate from the activation sites. The similarity of $K_d$ and deactivation rates in GIRK1$_{ΔdCT}$/2 and GIRK2 indicates that the activation sites in GIRK2 and GIRK1/2 have similar Gβγ affinities. If the anchor does not participate in channel opening, how does it increase the apparent affinity? We propose that this occurs by local enrichment of Gβγ around GIRK1/2 due to Gβγ recruitment[4], through kinetic scaffolding-like mechanisms[65-67], functionally equivalent to dynamic preassociation. The increased local Gβγ concentration,

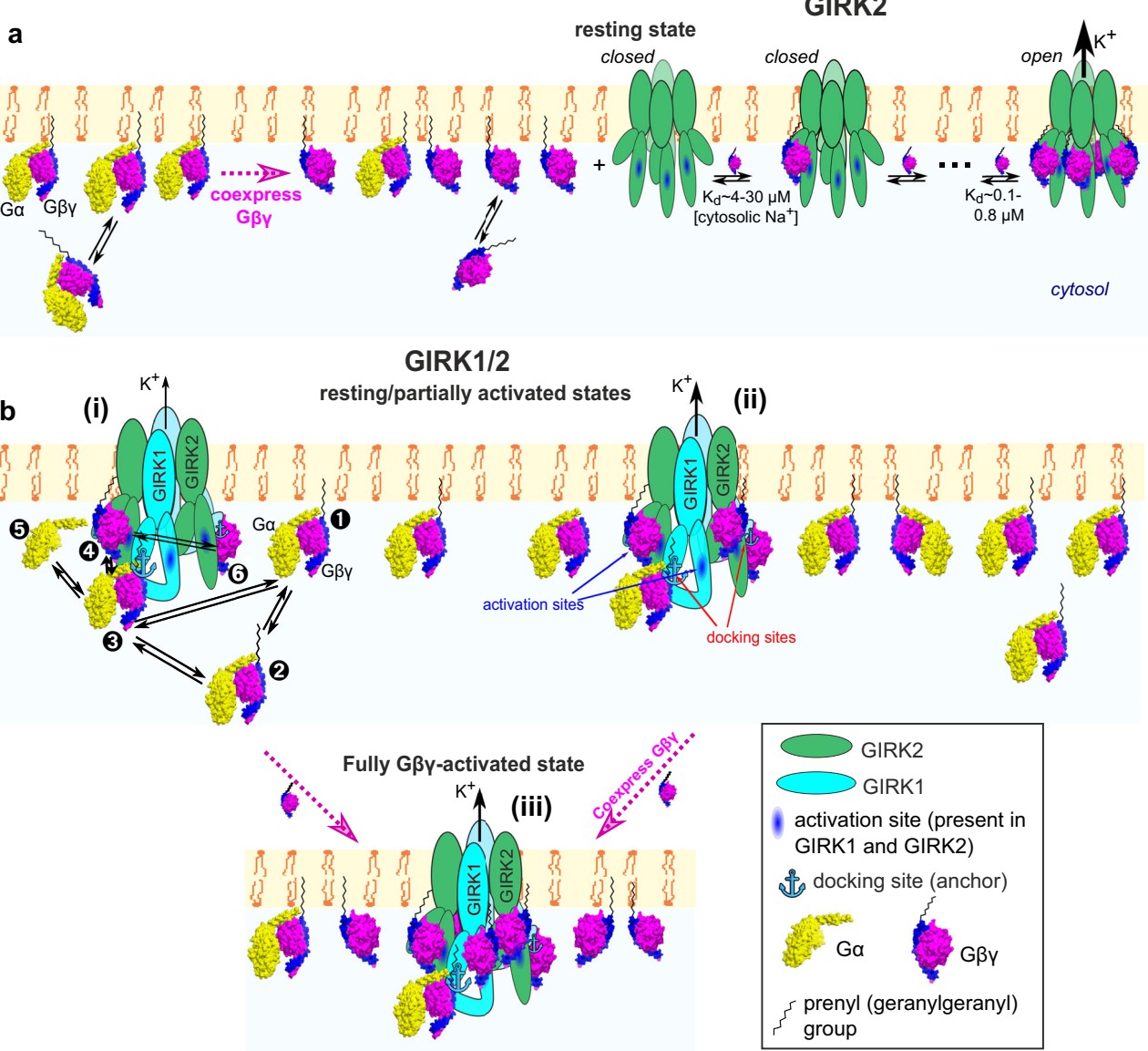

**Fig. 9 | Differences between GIRK2 and GIRK1/2 in their interaction and gating by Gβγ. a** GIRK2 homotetramer does not preassociate with Gβγ and has low $I_{basal}$. Channel opening requires the binding of four Gβγ. The affinity of first Gβγ binding is ~4-30 μM and increases with the binding of each additional Gβγ. **b** GIRK1/2 reversibly preassociates with Gβγ or Gαβγ due to two Gβγ-docking sites (anchors) formed by G1-dCT and NT (3,6) and opened following Gβγ binding to its activation sites (e.g., 4). In the "graded contribution" scenario shown, binding of even one Gβγ to an activation site induces opening, and $P_o$ as well as K⁺ flux are increased with each additional bound Gβγ. GIRK1/2 operates within a complex dynamic system that includes the channel and membrane-associated (1), cytosolic (2) and channel-bound Gαβγ and Gβγ, and free Gα$_{GDP}$ or Gα$_{GTP}$ (5). Gγ$_{prenyl}$ plays an important part in the emerging equilibrium by interacting with the PM or, alternatively, Gα, the anchor, and Gβ C-terminus (most of these interactions are not shown). The anchors attract Gβγ, leading to an enrichment of Gβγ and, potentially, Gαβγ in channel's microenvironment even in the absence of GPCR activation (basal states i, ii). Free Gβγ is in excess over Gαβγ because the presence of the anchor renders the channel with an overall higher affinity to Gβγ than Gα. Because of excess of free Gβγ, 1-3 out of the 4 activation sites of the GIRK1/2 tetramer are already occupied by Gβγ in basal state, $I_{basal}$ is high, and full activation (state iii) is achieved by binding of additional 1-3 Gβγ molecules.

in excess over Gα, leads to partial activation sites' occupation and high $I_{basal}$[33,35]. Moreover, the added Gβγ will bind to the subsequent (unoccupied) sites with higher affinity due to cooperativity, explaining the leftward shift in GIRK1/2's Gβγ dose-response curve. Added efficiency could arise if Gβγ's binding surfaces for docking and activation sites are non-overlapping, allowing the docked Gβγ to repeatedly contact the nearby activation site before Gβγ dissociation from the anchor. Mapping the anchor-Gβγ interface is a challenge for the future.

What is the role of Gα? Gα$_{i/o}$ interacts with GIRKs and was hypothesized to dock the Gα$_{i/o}$βγ heterotrimer to GIRKs[43,68,69]. However, the affinity of Gα to GIRK1 is lower than Gβγ[70]. Importantly, binding of Gα$_i$ to G1NC is enhanced by added Gβγ, suggesting that the heterotrimer is docked via Gβγ[4,34,71]. Both Gβγ-dependent Gα$_{i3}$-GIRK1 interactions and the speed and amplitude of $I_{evoked}$ are maximized when both G1-NT and G1-dCT are present[29,52,53,71], indicating that the NT-dCT anchor is involved in docking the heterotrimer (Fig. 9b). The stoichiometry of anchor-associated Gαβγ and Gβγ in cells likely varies with GIRK1/x density, constitutive GPCR activity, and other factors[4].

## Method

### Ethical approval and *Xenopus* oocytes handling

Experiments were approved by Tel Aviv University Institutional Animal Care and Use Committee (permits #01-20-083 and TAU-MD-IL-2411-174-3).

Maintenance and surgery of female frogs were done as

described[43]. Female frogs, aged 1.5-5 years, were purchased from Xenopus 1 Corp. (Dexter, MI, USA) and kept in dark colored plastic tanks at $20 \pm 2\,°C$ at 10/14-hour light-dark cycle. During surgeries, frogs were anesthetized with a 0.25% Tricaine methanesulfonate (MS-222, Sigma-Aldrich #886-86-2) solution, and parts of ovary were removed through a small abdominal incision. Oocytes were defolliculated with collagenase in $Ca^{2+}$ free ND96 solution (in mM: 96 NaCl, 2 KCl, 1 $MgCl_2$, 5 HEPES, pH 7.5). 2 hours later oocytes were washed with NDE solution (in mM: 96 NaCl, 2 KCl, 1 $MgCl_2$, 1 $CaCl_2$, 5 HEPES, 2.5 mM sodium pyruvate, 50 mg/ml gentamycin, pH 7.5) and left in NDE for 2-24 hours before injection. Oocytes were injected with 50 nl RNA using micro-injection pipette (Drummond Scientific, Broomall, PA, USA) and incubated at 20 °C for 72 hours for two-electrode voltage clamp, or 48-72 hours for single-channel patch clamp experiments.

### DNA constructs, RNA, antibodies

DNA constructs encoding the proteins used are summarized in Supplementary Table 9. Antibodies are described in relevant sections of the Methods and summarized in Supplementary Table 10. All antibodies were commercially available and validated by vendors. In each experiment involving detection of a protein in a sample derived from living cells, the specificity was validated by the absence of signal in cells not expressing the protein under study, and by comparison with purified recombinant proteins such as Gβ.

Gβγ stands for $Gβ_1γ_2$ throughout the paper. All DNA constructs used to produce RNA were inserted in vectors containing 5' and 3' untranslated sequences of *Xenopus* β-globin (pGEM-HE, pGEM-HJ or pBS-MXT)[71]. New constructs were prepared using standard PCR-based procedures (see Supplementary Data 2 for list of primers) and fully sequenced. We used the mouse isoform GIRK2A, which is 11 a.a. shorter than the longer isoform (mouse and human) not studied here, which includes a PDZ-binding consensus sequence at the dCT[72]. The truncated GIRK2 construct ($GIRK2_{trunk}$) was prepared by deleting a.a. 1–51 and 381–414 from the GIRK2A construct by PCR. $G2NC_{trunc}$ was prepared by deleting, from G2NC, of the same regions. YFP-$Gγ_{C68S}$ was prepared through single-nucleotide mutation of YFP-Gγ. Myristoylated $Gβ_1$ (myr-Gβ) was created by adding the myristoylation signal (the first 15 aa of Src added to the N terminus of $Gβ_1$)[32]. GIRK2-CFP was created by fusing $CFP_{A207K}$ to the CT of GIRK2 via a Ser-Arg linker. IRK1-YFP and IRK1-CFP were created by fusing $YFP_{A207K}$ and $CFP_{A207K}$, respectively, to the CT of IRK1 via a Lys-Leu linker[73]. N-terminally Split Venus labeled $Gβ_1$ (SpV-Gβ) and N-terminally Split Venus labeled $Gγ_2$ (SpV-Gγ)[59] were subcloned into pGEM-HJ. G1NdCT (the fused cytosolic G1-NT and G1-dCT), G1N(1-40) dCT (the first 40 a.a. of G1-NT fused to G1-dCT), G1N(40-84)dCT (the last 44 a.a. of G1-NT fused to G1-dCT), Sumo-G1NT and Sumo-G1dCT (Sumo fused to G1-NT or G1-dCT). In all cases the fusion was via the 8-a.a. linker, QSTASQST. The Sumo construct used here was a truncated version of human Sumo 2 protein (a.a. 3-95; PDB: 5ELU_B).

RNAs were transcribed in vitro as described[43]. The amounts of injected RNAs varied according to the experimental design. For whole-cell electrophysiology experiments we used, in ng/oocyte: 0.01-1 of GIRK1 or YFP-GIRK1, 0.2-10 GIRK2, 0.2-10 Gβ, 0.04-2.5 Gγ, 0.08-5 YFP-Gγ. Equal amounts of GIRK1 and GIRK2 RNAs were injected to express GIRK1/2 channels. In all experiments where several Gβγ expression levels were tested, the ratio of Gβ:Gγ RNA was kept constant: for Gβ:Gγ, the RNA ratio was 5:1 or 2.5:1, and for Gβ:YFP-Gγ the ratio was 2:1 or 2.5:1. For single channel patch clamp, the injected RNAs (in ng/oocyte) were: 0.005-0.01 IRK1-CFP; for GIRK2 alone, 0.02-0.05; for $GIRK1/2_{HA}$, GIRK1 0.01-0.02 of GIRK1 with 0.01-0.02 $GIRK2_{HA}$. In the experiments of Fig. 5, we injected, in ng/oocyte: GIRK2 alone, 0.2-0.5; GIRK1/2, 0.02-0.05 of GIRK1 and 0.01-0.025 of GIRK2; for GIRK1ΔdCT/2, 0.02-0.05 of GIRK1ΔdCT and 0.01-0.025 of GIRK2. In all patch clamp experiments with Gβγ-activated GIRKs, we injected 5 ng $Gβ_1$ and 1-2 ng $Gγ_2$ RNA, and 25-50 ng/oocyte of the GIRK5 antisense oligonucleotide[35] to prevent the formation of GIRK1/5 channels.

### Gβγ expression and purification

$His_6$-Gβγ and $His_6$-$Gβγ_{C68S}$ were purified essentially as described[74]. For full details see Supplementary Methods. $Gβ_1$ and $Gγ_2$ were expressed in *Trichoplusia ni* (*T.ni*) cells. The non-prenylated $His_6$-$Gβγ_{C68S}$ was extracted from the soluble fraction of the cells' homogenate and $His_6$-$Gβγ_{WT}$ from the membrane fraction, which ensures that the final purified protein is >95% prenylated[39]. Protein purity was analyzed using SDS-PAGE and by Western blot using anti-$Gβ_1$ and anti-His tag antibodies (Supplementary Table 10).

### Electrophysiology

Whole-cell GIRK currents were measured using standard two-electrode voltage clamp at 20–22 °C using GeneClamp 500B amplifier (Molecular Devices, Sunnyvale, CA, USA) and digitized using Axon Digidata 1440a using pCLAMP software (Molecular Devices). Agarose cushion microelectrodes were filled with 3 M KCl, with resistances of 0.1–1 MΩ[34]. GIRK currents were measured in either low-[$K^+$] solution ND96 (same as $Ca^{2+}$-free but with 1 mM $CaCl_2$) or high-K solution with 24 mM [$K$]$_{out}$ (in mM: 24 KCl, 72 NaCl, 1 $CaCl_2$, 1 $MgCl_2$ and 5 Hepes). In experiments of Fig. 1, to maximize GIRK2's $I_{basal}$, we used a 96 mM high-[$K$]$_{out}$ solution (in mM: 96 KCl, 2 NaCl, 1 $CaCl_2$, 1 $MgCl_2$ and 5 Hepes). Net GIRK currents ($I_{basal}$ and $I_{βγ}$) were determined by subtraction of currents recorded in presence of 1-2.5 mM $Ba^{2+}$ that blocked GIRK currents. The pH of all solutions was 7.5–7.6. Cell-attached patch clamp recordings were performed at 20–23 °C, using borosilicate glass pipettes with resistances of 1.5–3.5 MΩ. The electrode solution contained (in mM): 146 KCl, 2 NaCl, 1 $CaCl_2$, 1 $MgCl_2$, 10 Hepes and 1 $GdCl_3$ (pH 7.6). Bath solution contained (in mM): 146 KCl, 2 $MgCl_2$, 6 NaCl, 10 Hepes and 1 EGTA (pH 7.6). Block of stretch-activated channels by $GdCl_3$ was confirmed by recording currents at +80 mV. Single channel currents were recorded at −80 mV in cell-attached patches with the Axopatch 200B amplifier (Molecular Devices) at −80 mV, filtered at 2 or 5 kHz and sampled at 10 or 25 kHz.

### Giant membrane patches (GMPs)

GMPs were prepared and imaged as described[59]. Oocytes were devitellinized using tweezers in hypertonic solution (in mM: 6 NaCl, 150 KCl, 4 MgCl2, 10 Hepes, pH 7.6). The devitellinized oocytes were transferred onto a Thermanox™ coverslip (Nunc, Roskilde, Denmark) and immersed in $Ca^{2+}$-free ND96 solution with their black hemisphere facing the coverslip, for 30–45 min. The oocytes were then suctioned using a Pasteur pipette, leaving a GMP attached to the coverslip, with the cytosolic part facing the medium. The coverslip was washed thoroughly with fresh ND96 solution, and fixated using 4% formaldehyde for 30 min. Fixated GMPs were immunostained in 5% milk in PBS and non-specific binding was blocked with Donkey IgG 1:200 (Jackson ImmunoResearch, West Grove, PA, USA). Primary rabbit anti-Gβ (1:200; Santa Cruz, SC-378 or GeneTex, GTX114442) was applied for 45 min at 37 °C either alone or with blocking peptide supplied with the antibody. Then DyLight549 or DyLight® 650-conjugated anti-rabbit secondary antibodies (KPL) were applied at 1:300 dilution for 30 min at 37 °C, washed with PBS and mounted on a slide for visualization. Immunostained slides were kept at 4 °C in the dark.

### Preparation of whole oocyte lysates and separated plasma membranes for pull-down and WB

Lysates from whole nucleus-free oocytes were prepared as described[43]. 6–10 oocytes were homogenized on ice (20 mM Tris, pH 7.4, 5 mM EGTA, 5 mM EDTA, 100 mM NaCl) containing Roche Complete Protease Inhibitors Cocktail (Merck 11697498001, 1 tablet/100 ml, pH=7.5), 6 µl buffer/oocyte. Nucleus and yolk were removed by centrifugation (1000×g, 15 minutes, 4 °C). Supernatant was stored in aliquots corresponding to two oocytes at −80 °C. Manually separated oocytes' PMs for qWB have been prepared as described[35]. PMs together with the vitelline membranes (extracellular collagen-like matrix)

were manually separated from the rest of the oocyte ("cytosol") with fine forceps, after a 5–15 min incubation in a low osmolarity solution (5 mM NaCl, 5 mM HEPES, and protease inhibitor as above. PMs of ~20 oocytes were pooled for each sample (lane on gel).

### Pull-down assays, autoradiograms and WB
Pull-down binding experiments were performed as described[33]. For full description see Supplementary Methods. Briefly, pull-down was done with in vitro translated (*ivt*) [³⁵S]methionine-labelled proteins prepared in rabbit reticulocyte lysate, or unlabeled proteins from whole-oocyte lysates, with ~2 μg of either purified His-Gβγ$_{WT}$ or purified His-Gβγ$_{C68S}$, in 300 μl of the incubation buffer (in mM: 150 KCl, 50 Tris, 0.6 MgCl$_2$, 1 EDTA, 0.1% Lubrol or 0.5% CHAPS; pH 7.4), followed by 60 min incubation and then addition of Ni-NTA Resin affinity beads and imidazole for 30 min. After repetitive washing, His-Gβγ and bound material were eluted with 250 mM imidazole and subjected to SDS-PAGE. [³⁵S] Methionine-labeled proteins were detected by autoradiography and unlabeled proteins by WB with the appropriate antibodies, and quantified with ImageJ/Fiji (https://imagej.net/software/fiji/). For expressed Gβ, endogenous Gβ signal from oocytes expressing the channel alone was subtracted from the total signal.

### Confocal imaging
Confocal imaging and analysis were performed as described[73]. See Supplementary Methods for details. Live oocytes were imaged at their animal hemisphere. Giant membrane patches were imaged at their edges to show both the membrane and the background. Net signals were calculated by subtracting the average net signal from uninjected (native) oocytes of the same experiment.

### Peptide spot array
Peptide arrays were generated by automatic SPOT synthesis and blotted on a Whatman membrane[75]. N-terminal and C-terminal parts of GIRK1 and GIRK2 were spot-synthesized as 25-mer peptides overlapping sequences, shifted by 5 a.a. along the sequence, using Auto-Spot Robot ASS 222 (Intavis Bioanalytical Instruments, Cologne, Germany). The peptides were designed according to human GIRK2 (NCBI: NM_002240.5) (NT: a.a. 1-93, CT: a.a. 193-423) and rat GIRK1 (NCBI: NP_113798.1) (NT: a.a. 1-84, CT: a.a. 183-501). The interaction with spot-synthesized peptides was investigated by an overlay assay. Following blocking of 1 hour at room temperature with 5% BSA in 20 mM Tris and 150 mM NaCl with 0.1% Tween-20 (TBST), 0.016–0.16 μM purified His-Gβγ were incubated with the immobilized peptide-dots, overnight at 4 °C. His-Gβγ was detected by anti-GNB1 antibody (GTX114442) at 1:500 or 1:1000 dilution, and anti-rabbit HRP-coupled secondary antibody (1:40000) incubated with 5% BSA/TBST, and the membrane was imaged using Fusion FX7, as for Western blotting.

### Electrophysiological data analysis and surface density calibration
Whole-cell and single-channel data were analyzed using Clampex and Clampfit (pCLAMP suite, Molecular Devices, Sunnyvale, CA, USA). In oocytes expressing the m2 receptor, the fold activation by agonist, $R_a$, was measured in each cell and defined as

$$R_a = I_{total}/I_{basal} \qquad (1)$$

where $I_{total} = I_{basal} + I_{evoked}$. $R_a = 1$ when there is no response to agonist.
The fold activation by Gβγ, $R_{\beta\gamma}$, was defined as

$$R_{\beta\gamma} = I_{\beta\gamma}/\bar{I}_{basal} \qquad (2)$$

where $I_{\beta\gamma}$ is the net GIRK current in a Gβγ-expressing oocyte, and $\bar{I}_{basal}$ is the average GIRK current in oocytes of control group, that express only the channel, from the same experiment[34].

Single channel amplitudes were calculated from Gaussian fits of all-points histograms of 30–90 s segments of the record. The open channel probability ($P_o$) was estimated from 1–5 min segments of 4–20 min recordings from patches containing one to three channels using a standard 50% idealization criterion[35].

The PM density of functional channels was determined from the whole-cell current, I, using the classical equation[46]

$$I = N_{ch} \cdot i_{single} \cdot Po \qquad (3)$$

where $N_{ch}$ is the total number of channels in the cell, $i_{single}$ is the single-channel current and $P_o$ is the open probability. $P_o$ and $i_{single}$ for Gβγ-activated GIRK1/2 are known, and for GIRK2, GIRK1/2$_{HA}$ and IRK1-xFP we determined them here (Supplementary Fig. 3, Supplementary Table 3). The surface density, in channels/μm² (μm⁻²) was calculated by dividing $N_{ch}$ by the membrane surface area of the oocyte[76], $2 \cdot 10^7$ μm². Protein surface densities were converted to concentrations using the standard procedure based on a submembrane interaction space 10 nm deep. $i_{single}$ was measured in cell-attached patches in 146 mM [K⁺]$_{out}$; whole-cell currents were measured in 24 mM [K⁺]$_{out}$. The amplitude translation factor for these solutions was 4.63. The conversion factor from surface density to sub-PM space concentration was 1 μm⁻² = 0.166 μM[35]. In calculating the surface density of channel-attached YFP (two for YFP-GIRK1/2 and four for IRK1-YFP), we assumed similar levels of fluorescence maturation of channel- and Gβ-attached YFP molecules, therefore no correction for such maturation was made. For CF calibrations with YFP-GIRK1/2 or IRK1-YFP, the linear fit included the zero-fluorescence point (with no expressed channels).

In the analysis of Gβγ dose-response data in intact oocytes, we assumed that the PM level of the GIRK2 channels was not significantly altered by Gβγ, as shown previously[34,59] and confirmed for CFP-GIRK2 (Supplementary Fig. 6). In one experiment we monitored GIRK2HA and observed changes at different doses of Gβγ, and corrected the currents accordingly (Supplementary Table 6). Similarly, coexpression of Gβγ causes no significant changes in PM levels of GIRK1/2 up to 2 ng RNA of Gβ[59]. In most experiments, the maximal GIRK1/2 current was observed already with 1 or 2 ng Gβ RNA. With 5 ng Gβ RNA, a 20–30% decrease in channel expression is occasionally seen[59]. No correction for this potential change has been made.

### Modeling, simulation and curve fitting for Gβγ dose-response data
Standard fitting for Gβγ-GIRK dose-response curves with Hill or modified WTM models was done assuming that, in the absence of GPCR simulation, the endogenous G proteins are in the form of heterotrimers. Data were fitted to Hill equation in the following form:

$$I_{GIRK} = (1-c)I_{max}x^{nH}/(x^{nH} + Kd^{nH}) + cI_{max} \qquad (4)$$

where $x$ is the concentration of coexpressed Gβγ ([Gβγ]), $I_{GIRK}$ is GIRK current, $I_{max}$ is the maximal GIRK current at saturating concentrations of coexpressed Gβγ, $n_H$ (or nH) is the Hill coefficient, $c$ is a constant component corresponding to $I_{basal}$;
or a modified WTM model[15] with the addition of a constant component c:

$$I_{GIRK} = ((1-c)I_{max}x^4/(K_d^4\mu^6 + 4K_d^3\mu^6x + 6K_d^2\mu^5x^2 + 4K_d\mu^3x^3 + x^4))$$
$$+ cI_{max}$$

$$(5)$$

where $x$, $c$ and $I_{max}$ have the same meaning as in Eq. 4, $K_d$ is the dissociation constant of the first Gβγ binding to the one of the four sites in GIRK molecule, μ is the cooperativity factor for each successive Gβγ binding[24] for the specific case of a constant $Na^+$ concentration[15]. In whole-cell of cell-attached recordings from intact *Xenopus* oocytes, both intracellular $Na^+$ and the membrane $PIP_2$ can be assumed constant during the experiment. Therefore, in most WTM model fits, we utilized a constant cooperativity factor μ=0.3[15] or μ=0.44 (from Fig. 3). In two experiments with GIRK2 we were able to obtain independent estimates of μ from fit, which were 0.44 and 0.62 (Supplementary Table 6, "free μ").

To simulate Gβγ activation of GIRK1/2, we tested four kinetic schemes (models) (Supplementary Fig. 8a). First, we calculated the basal available Gβγ and Gα from the experimentally observed $I_{basal}$[14,35]. For simulation, we constructed systems of differential equations based on these schemes and solved them numerically. See Supplementary Methods for details.

### Molecular dynamics simulations

All MD analyses were performed with publicly available software. Systems were built using CHARMM-GUI (accessed July 2024). Simulations were run in GROMACS 2022.3. The Martini Elnedyn22p coarse-grained force field was applied, and atomistic refinements used the Amber14SB force field. Simulations were run using GROMACS 2022.3. Molecular graphics and analyses were performed with VMD 1.9.4a12 (December 2017); trajectories were analyzed with MDAnalysis 0.20.1 implemented in Python 3.7.4 and VMD 1.9.4a12. We also used Python 3.7.4 routines NumPy 1.21.6, pandas 1.3.5, matplotlib 3.1.3, seaborn 0.11.1. Full details and references related to MD simulation are in Supplementary Methods and Supplementary Tables 11–14.

Primary structures of G1NC and G2NC were generated by fusing the NT and CT of human GIRK1 and human GIRK2, respectively (Fig. 8). The heatmaps in Fig. 8 show G412 as the last a.a., which corresponds to G414 of mGIRK2 (Supplementary Fig. 9). Additionally, Gβγ units were incorporated into the sequences. Truncated constructs were the same as G1NC$_{ΔdCT}$ and G2NC$_{trunc}$ used in biochemical experiments (Fig. 1e).

### Statistical analysis

Statistical analysis was performed using GraphPad Prism (GraphPad, La Jolla, CA, USA). For normally distributed data (by Shapiro-Wilk test), pairwise comparison was done by t-test and multiple comparisons by one-way ANOVA, and data were presented as bar graphs with individual data points and mean ± SEM (except if non-normally distributed data were presented on the same panel, in which case box plots were shown). If the data did not pass the normal distribution test, they were analyzed using Mann-Whitney (pairwise) and Kruskal-Wallis nonparametric ANOVA tests, and data were presented as box plots and individual data points. The boxes represent the 25th and 75th percentiles, the whiskers show the smallest and maximal values, and the horizontal line represents the median. Statistical analysis for differences between dose-response curves for two different GIRK compositions was done on WTM model fits of normalized dose-response data from individual oocytes for two fits (as in Fig. 4b, d), as well on three fits (details in Supplementary Fig. 7).

### Graphics

Structures of GIRK2, Gα and Gβγ were drawn with PyMOL (Schrodinger LLC). All final figures were produced with Inkscape (inkscape.org). Molecular graphics and analyses were performed with VMD 1.9.4a12 (December 2017).

### Reporting summary

Further information on research design is available in the Nature Portfolio Reporting Summary linked to this article.

### Data availability

All data are presented in figures and tables in the main paper and in Supplementary Material. Source Data are provided with this paper. The molecular dynamics (MD) simulation data generated in this study have been deposited in Zenodo (https://zenodo.org/records/17117723)[77] and are publicly available without restrictions. All materials created in this paper, such as DNA constructs, are fully available upon request. The source data underlying Fig. 1b–d, f, g, Fig. 2a, b, c–f, h, i, Fig. 3, Fig. 4a–d, g, Fig. 5d, e, Fig. 7, Supplementary Figs. 1b, c, e, f, 3c, 4e, f, 6a–d, 7c, 11, and Supplementary Tables 3, 4 are provided as Source Data file. Previously published structures referred to in this study are available from the Protein Data Bank under accession codes 5ELU and 1GP2. Source data are provided with this paper.

### Code availability

No custom code was used in this study. All analyses were performed with publicly available software as described in the Methods and Supplementary Methods.

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

## Acknowledgements

This work was supported by grants 1282/18 and 581/22 from Israel Science Foundation (ND), KL1415/14-1 from the Deutsche Forschungsgemeinschaft (EK), R35GM145921 from the National Institute of General Medical Science, National Institutes of Health, W1232 from the Austrian Science Fund (A.S.W., T.F.). The computational results presented have been achieved in part using the Vienna Scientific Cluster (V.S.C.).

## Author contributions

N.D., D.Y., A.S.W.–conceptualization. R.H., T.K.R., B.S., U.K., T.F., C.W.D., Y.H., J.A.H., A.S.W., D.Y., N.D.–experimental design. R.H., T.K.R., B.S., P.H., U.K., T.F., G.T., V.T., H.R.P., A.S.W., D.Y., N.D.–investigation. R.H., G.T., O.C.H., D.R.T., K.Z., C.W.D., E.K., J.A.H.–resources. R.H., T.K.R., P.H., U.K., T.F., A.S.W., N.D.–data curation. E.K., Y.H., J.A.H., A.S.W., D.Y., N.D.–supervision. C.W.D., E.K., Y.H., J.A.H., A.S.W., N.D.–funding acquisition. R.H., A.S.W., D.Y., N.D.–writing the initial version of the paper. R.H., H.R.P., U.K., E.K., C.W.D., Y.H., J.A.H., A.S.W., D.Y., N.D.—review and editing. All authors read and confirmed the paper.

## Competing interests

The authors declare no competing interests.

## Additional information

[1]Department of Physiology and Pharmacology, Faculty of Health and Medical Sciences, Tel Aviv University, Tel Aviv, Israel. [2]Department of Pharmaceutical Sciences, Division of Pharmacology and Toxicology, University of Vienna, Josef-Holaubek-Platz 2, Vienna, Austria. [3]Department of Biochemistry & Molecular Biology, School of Neurobiology, Biochemistry and Biophysics, George S. Weiss Faculty of Life Sciences, Tel Aviv University, Tel Aviv, Israel. [4]Max-Delbrück-Center for Molecular Medicine in the Helmholtz Association (MDC), Berlin, Germany. [5]Department of Integrative Biology and Pharmacology, University of Texas Health Science, Center, Houston, Texas, USA. [6]DZHK (German Centre for Cardiovascular Research), partner site Berlin, Berlin, Germany. [7]Sagol School of Neuroscience, Tel Aviv University, Tel Aviv, Israel. [8]The Adelson School of Medicine, Ariel University, Ariel, Israel. [9]Present address: Department of Neuroscience, Faculty of Health and Medical Sciences, University of Copenhagen, Blegdamsvej 3B, 2200 Copenhagen, Denmark.
✉e-mail: anna.stary@univie.ac.at; danielya@ariel.ac.il; dascaln@tauex.tau.ac.il

