## [Transparent Peer Review file · Nature Communications]

Live-cell quantitative monitoring reveals distinct, high-affinity G β γ regulations of GIRK2 and GIRK1/2 channels

Corresponding Author: Professor Nathan Dascal

Version 0:

Reviewer comments:

Reviewer #1

(Remarks to the Author)

The authors use a multi-faceted approach involving protein biochemistry, electrophysiology, and molecular dynamic simulations to address a complex problem – the nature of the Gbetagamma regulation of the GIRK channel. The main goal is to address inconsistencies in the literature regarding the affinity of the Gbetagamma-GIRK interaction. The authors present evidence that prenylation of the Ggamma subunit, a known determinant of membrane localization of Gbetagamma, enhances the affinity of the interaction. Their data also suggest that GIRK1/GIRK2 channel, a common channel subtype in neurons, presents a docking site for prenylated Gbetagamma that consists of N- and C-terminal GIRK1 segments and that this is distinct from Gbetagamma “activation” domain.

The authors should be commended for a rigorous and thoughtful study. The main concern relates to novelty/impact of the study. The importance of prenylation in regulating the Gbetagamma-GIRK interaction (and other Gbetagamma interactions) and its affinity has been appreciated for some time. The few studies reporting low affinities involved unprenylated Gbetagamma and either relatively small GIRK subunit fragments or truncated GIRK2 homomeric channels. The role of GIRK1 and its intracellular domains (and combined N/C-terminal domain) in enhancing Gbetagamma-channel interactions and function is also appreciated. In this context, the study does not offer many new insights but rather (impressive) refinement of conclusions from published studies.

Other limitations of the study:

1) The authors used purified prenylated Ggamma2 but no information is provided about the degree of prenylation; are 100% of Ggamma2 prenylated? Relatedly, is not clear whether the Ggamma2 subunit undergoes efficient prenylation in the oocyte or not. Overall, verification or quantification of Ggamma2 prenylation in these studies would be helpful.

2) Does the type of prenylation (e.g. farnesyl or geranylgeranyl) impact binding to the Gbetagamma anchor? It is not clear whether the anchor presents a specific binding pocket for Gbetagamma that might be sensitive to different lipid modifications or combinations of Gbeta and Ggamma.

Reviewer #2

(Remarks to the Author)

see attached

Reviewer #3

(Remarks to the Author)

Reviewer #4

(Remarks to the Author)

Handklo-Jamal et al. addressed a long-standing question in GIRK channel regulation by integrating peptide mapping with coarse-grained (CG) molecular dynamics (MD) simulations to propose a cooperative “Gβγ-anchor” mechanism involving the GIRK1 N-terminus, distal C-terminus, and geranylgeranyl tail of Gy. The combined use of structural modeling, simulations, and biochemical validation is both original and compelling. This work is of clear interest to the ion channel and signalling communities, particularly those studying GIRK regulation, GPCR–G protein–effector coupling, and the effects of lipid modification. This builds upon the foundational GIRK–Gβγ structures and expands our understanding of previously unresolved regulatory regions. Most conclusions are reasonably supported by the data; however, several important concerns remain:

1. All CG simulations were conducted without a membrane, despite the highly lipophilic nature of the geranylgeranyl moiety. In Supplementary Fig. S10, the tail frequently associates with GIRK surfaces that would ordinarily be embedded within the bilayer. To substantiate the proposed mechanism, the authors should either perform additional simulations incorporating a membrane (e.g. three CG replicas) or explicitly acknowledge this limitation and adjust their mechanistic claims accordingly.
2. While CG simulations accelerate dynamics, a microsecond may still be insufficient to capture slower cooperative rearrangements, especially in multimeric membrane-associated systems. The authors should either extend their simulations to the multi-microsecond range or provide a more detailed analysis of the contact profiles across independent 1-μs replicas, including evidence of convergence.
3. The reported interaction percentages and reductions are presented without uncertainties or significance testing. To strengthen their conclusions, the authors should report the mean ± SEM across replicas and apply appropriate statistical tests when claiming significant differences between constructs.
4. To validate the stability and resolution of the proposed interaction beyond the CG level, the authors should perform at least one atomistic MD refinement of the interface.
5. The use of AlphaFold to model full-length GIRK subunits is an ambitious approach. However, the reliability of these predictions in multimeric complexes with flexible termini and membrane-associated regions should be discussed in greater depth, including any limitations and confidence levels of the predicted structures.

Version 1:

Reviewer comments:

Reviewer #1

(Remarks to the Author)

I appreciate the thorough and thoughtful response of the authors to the concerns raised by the reviewers of the original manuscript. I think the manuscript as it stands now represents an important addition to the literature on this canonical Gbetagamma/effector interaction. I have no further recommendations for manuscript improvement.

Reviewer #2

(Remarks to the Author)

The revised manuscript is much improved and I recommend it be published.

Reviewer #3

(Remarks to the Author)

Reviewer #4

(Remarks to the Author)

The authors have comprehensively addressed my previous concerns. They expanded coarse-grained simulations to include membrane-embedded systems, conducted longer trajectories reaching multi-microsecond timescales, incorporated atomistic refinements, and provided statistical analyses with appropriate uncertainties. Furthermore, they clarified the limitations of AlphaFold predictions and supported their mechanistic model with new experimental data. These substantial additions enhance the study's robustness.

Live-cell quantitative monitoring reveals distinct, high-affinity G $\beta\gamma$ regulations of GIRK2 and GIRK1/2 channels

Summary: This manuscript reports the importance and function of G γ prenylation for affinity and subsequent activation of GIRK channels, and describes a G $\beta\gamma$ anchor located in the N-terminal and distal C-terminal region of G1 GIRK channels. The authors investigate these biochemical features of G γ and GIRK channels in *Xenopus* oocytes. This allows the authors to address a discrepancy in the field regarding G $\beta\gamma$ affinity for GIRK channels using prenylated or non-prenylated G $\beta\gamma$. Assessing current in these *Xenopus* oocytes with prenylated and non-prenylated G $\beta\gamma$ shows significant affinity enhancements with prenylated G $\beta\gamma$. The authors also utilize fluorophore tagged GIRK and G γ to obtain a K_d of G $\beta\gamma$ for GIRK 1/2 and GIRK2. They then define the G $\beta\gamma$ anchor on GIRK1 subunits via dot blot analysis and pull down quantitative western blotting to show deletion of G1-NT and G1-dCT reduces affinity for G $\beta\gamma$. The authors also support their claims through molecular dynamic modeling that agrees with their biochemical conclusions.

This manuscript describes important biochemical interactions between G $\beta\gamma$ and GIRK channels that should be considered whenever investigating this interaction. However, there are some substantive issues that the authors should consider:

Major Critiques:

- In the section of the results titled “**G1-NT and G1-dCT form a G $\beta\gamma$ -binding site and contribute to channel’s interaction with G γ ’s prenylation tail, G γ _{prenyl}” claims are made that the prenylation of G γ is critical for the interaction of G $\beta\gamma$ with the G1-NT and G1-dCT fusion protein, as well as specifically the dCT, as its deletion causes less binding. While this was supported with MD simulations, this was not confirmed by a pull-down assay with non-prenylated G $\beta\gamma$. If this claim is true, non-prenylated G $\beta\gamma$ should display a lower yield of G1NC and G2NC in that assay. While *in-silico* methods are useful at identifying potential interactions, these should be confirmed *in-vitro* to make this claim.**
- Defining the G $\beta\gamma$ anchor within G1-NT and G1-dCT was done via truncation of the protein leading to a decrease in the G $\beta\gamma$ binding which is supported. However, truncation of channel subunits in this manner brings up concern about the biological implication of these deletions. Primary sequence length is known to be critical for proper protein folding. In this manuscript, there is no validation that the protein is in fact folded properly. Alanine mutation in these regions displaying the same results as the truncation would significantly strengthen the support of these claims.

Minor Critiques:

- In the first sentence of the third paragraph of the introduction, the G $\alpha\beta\gamma$ should be specified as G $\alpha_{i/o}\beta\gamma$.

Reply to reviewers

We thank the Reviewers for helpful and constructive comments and suggestions. We have addressed all the comments and provide a point-by-point response to them. Main changes made in the MS text and Figure legends are highlighted in blue font. In addition, we are attaching a combined file of the main part with supplementary results, containing all figures with their legends (like in the first submission), with changes highlighted in the same way. We believe that it may be easier for the Reviewers to track all changes, including the ones in the Supplementary Material, in this way. To further facilitate easy tracking of changes, we are attaching a small file, “List of edits and additions”, describing the changes and additions made in the paper. This list is also added in the beginning of our Reply to Reviewers (below).

List of edits and additions in the revised MS.

For Reviewers’ convenience, we have attached a combined file (MS + Supplementary Material) in which edits and additions, except minor grammar corrections (or rephrasing done in order to keep within the 5000 word limit) are highlighted in blue font.

New Figures and Tables added.

- Fig. 8: all data were updated for the new 5 μ s runs (instead of 1 μ s). Results of MS simulation with membrane were added in 8d (bottom).
- Supplementary Fig. 1: new panels b, c were added (reporting new experiments, WB of YFP-G β γ and YFP-G β γ _{C68S})
- Supplementary Fig. 2 legend: more details about statistical analysis, which were mentioned in the main text, were added.
- Supplementary Fig. 4e: we removed one panel (2d from right) as we realized that there was a double presentation of one experiment for 6Gly-YFP G γ , which was also presented as part of summary from two experiments in the right panel of this figure (which is the one presented now).
- Supplementary Fig. 10: new images of all simulation systems, including G1NC with POPC membrane and G1NC all-atom backmapped for AA simulations. New panel c comparing G γ _{prenyl} interactions in all simulation systems. Previous panel c (a table showing interactions of G γ _{prenyl} with G β residues) was removed and replaced by the new Supplementary Table 13.
- New Supplementary Fig. 11 reporting the new experiments showing the difference in G1NdCT binding to prenylated vs. non-prenylated G β γ .
- New Supplementary Table 11 shows the results of the new experiment testing the interaction of G1NdCT with G β γ _{WT} and G β γ _{C68S}.
- Correction in suppl. table 4: data on the endogenous G β γ in oocytes expressing GIRK2 were updated with the inclusion of two additional experiments, changing the G β surface density from 35 to 28 μ m⁻² and N from 3 to 5.
- Supplementary Table 10: two antibodies used in the new experiments were added.
- New Supplementary Tables 11-14: details of all MD simulations.
- New Supplementary MD figures collection showing analysis of all simulation in all systems in graphical form.

Changes in main and Supplementary text:

- **Introduction:** the last 4 sentences summarize the findings and conclusions, replacing the removed Summary (previously at the end of Discussion).

- **Results**

- 1. Lipid modification of G γ is essential for GIRK activation and important for GIRK-G $\beta\gamma$ interaction:**

- ◊ New experiments comparing **expression** of YFP-G γ and YFP-G γ_{C68S} in whole oocytes (Supplementary Fig. 1b,c) – paragraph 2.

- 2. G1-NT and G1-dCT form a G $\beta\gamma$ -binding site and contribute to channel's interaction with G γ 's prenylation tail, G γ_{prenyl} :**

- ◊ New CG 5 μ s simulations and AA simulations described – paragraphs 3, 4.
 - ◊ New experiment comparing G1NdCT binding to prenylated vs. non-prenylated G $\beta\gamma$ – paragraph 4.

- **Discussion:** additions are described in Replies to Reviewers.

- **Methods:**

- ◊ **G $\beta\gamma$ expression and purification** are described in brief with an emphasis on differences between G $\beta\gamma_{WT}$ and G $\beta\gamma_{C68S}$. Full protocols were extended and moved to Supplementary Methods.

- ◊ **Preparation of whole oocyte lysates** for the new experiments described.

- ◊ **Pull-down assays and confocal imaging protocols** reported in brief, full protocols were extended and moved to Supplementary Methods.

- **Supplementary Methods:** we added full extended protocols for G $\beta\gamma$ expression and purification, pulldown from reticulocyte and whole-oocyte lysates

Reply to reviewer #1:

The authors use a multi-faceted approach involving protein biochemistry, electrophysiology, and molecular dynamic simulations to address a complex problem – the nature of the Gbetagamma regulation of the GIRK channel. The main goal is to address inconsistencies in the literature regarding the affinity of the Gbetagamma-GIRK interaction. The authors present evidence that prenylation of the Ggamma subunit, a known determinant of membrane localization of Gbetagamma, enhances the affinity of the interaction. Their data also suggest that GIRK1/GIRK2 channel, a common channel subtype in neurons, presents a docking site for prenylated Gbetagamma that consists of N- and C-terminal GIRK1 segments and that this is distinct from Gbetagamma “activation” domain.

The authors should be commended for a rigorous and thoughtful study. The main concern relates to novelty/impact of the study. The importance of prenylation in regulating the Gbetagamma-GIRK interaction (and other Gbetagamma interactions) and its affinity has been appreciated for some time. The few studies reporting low affinities involved unprenylated Gbetagamma and either relatively small GIRK subunit fragments or truncated GIRK2 homomeric channels. The role of GIRK1 and its intracellular domains (and combined N/C-terminal domain) in enhancing Gbetagamma-channel interactions and function is also appreciated. In this context, the study does not offer many new insights but rather (impressive) refinement of conclusions from published studies.

We would like to argue with two of the suppositions above. First, the importance of prenylation in regulating the G $\beta\gamma$ -GIRK interaction has been greatly underappreciated, despite our early studies showing that it was essential for GIRK activation by G $\beta\gamma$ (Schreibmayer et al., 1996, 10.1038/380624a0) (although at that time we could not distinguish between the two roles of prenylation, PM insertion vs. interaction with the effector). We thoroughly discuss in the introduction the NMR and model bilayer studies by Shimada's and Mackinnon's groups that have been most influential in the last decade, have been widely cited, and have dictated the prevailing vision of how GIRKs are regulated by G $\beta\gamma$. These studies, while having contributed highly important and comprehensive insights into GIRK mechanisms, completely disregarded the possible role of Gy prenylation in effector interaction, beyond membrane insertion alone (and most of their papers also did not mention the latter in relation to other interactors of G $\beta\gamma$). Accordingly, these studies reported very low GIRK-G $\beta\gamma$ affinities. The controversy regarding GIRK-G $\beta\gamma$ affinity that we discuss in the introduction is real, and we believe that our work resolves it.

Second, whereas the importance of both N- and C-terminal cytosolic domains for GIRK-G $\beta\gamma$ interactions has been well studied, the composition of the high-affinity docking site (“anchor”) in GIRK1 was completely unknown. In fact, the mere existence of the anchor, which we have proposed in our previous works, has not been widely accepted. The demonstration and composition of the existence of G1 NT-dCT module as a high-affinity binding site in GIRK1 is a novel finding.

Other limitations of the study:

1) The authors used purified prenylated Ggamma2 but no information is provided about the degree of prenylation; are 100% of Ggamma2 prenylated? Relatedly, is not clear whether the Ggamma2 subunit undergoes efficient prenylation in the oocyte or not. Overall, verification or quantification of Ggamma2 prenylation in these studies would be helpful.

Thank you for these very important suggestions. We have relied on the well-established notion of Gy prenylation and its role in Gβγ function (we have cited several papers on this issue, e.g. the review by Escriba et al. 2007). However, we agree that there were omissions in the establishment of prenylation of Gy in our own experiments.

For purified recombinant Gβγ, we have previously cited the papers describing the method of production of prenylated Gβγ but did not thoroughly describe the complete protocols used by us. This is now corrected; full protocols are presented in Supplementary Methods. The main difference is that the prenylated Gβγ is extracted from the membrane fraction, where it is highly enriched and ensures >95% prenylated product after elution (Iñiguez-Lluhi et al., 1992). In contrast, Gβγ_{C68S} is extracted from the soluble fraction. These important differences are also emphasized now in the main Methods section: “The non-prenylated His₆-Gβγ_{C68S} was extracted from the soluble fraction of the cells’ homogenate and His₆-Gβγ_{WT} from the membrane fraction, which ensures that the final purified protein is >95% prenylated³⁹”. We apologize for the omission on this very important matter.

Regarding Gy prenylation in oocytes: we agree that the efficiency of prenylation in these cells has not been established. We have verified that Gβγ_{C68S} does not translocate to the plasma membrane (PM), in contrast to Gβγ_{WT}, which is clearly enriched, as shown in Supplementary Fig. 1. However, we realized that the absence of Gβγ_{C68S} in the PM could have occurred because it was not well expressed in the oocytes. We have performed additional experiments comparing YFP-Gβγ_{WT} vs. YFP-Gβγ_{C68S} expression in whole-oocytes (YFP-fused constructs were used to enable detection with the high-quality GFP antibody; we are not aware of good antibodies for Gy₂). We observed similar expression of both Gβ and Gy with either or Gy_{C68S} (the new Supplementary Fig. 1b,c). Accordingly, we have discussed this issue in the Results, paragraph 2: “We verified that N-terminally labeled YFP-Gy and YFP-Gy_{C68S} were comparably expressed in whole oocytes and support the expression of Gβ (Supplementary Fig. 1b,c). To assess PM localization, we immunostained Gβ in excised giant membrane patches (GMP)^{32, 43} using wild-type (WT) Gβ or an N-terminally myristoylated Gβ (myr-Gβ). Only WT Gy (Gy_{WT}), but not Gy_{C68S}, supported GIRK2 activation and, correspondingly, PM enrichment of Gβ_{WT} and myr-Gβ (Supplementary Fig. 1d-f)... These results confirm proper prenylation of Gy in oocytes, which is essential for PM attachment of Gβγ and GIRK2 activation;...”.

Taken together, these results verify that only prenylated Gβγ, produced by the oocyte, is reaching the PM; and since our protein calibration and measurement procedures are monitoring only the PM and PM-attached proteins, only prenylated Gy is measured. Accordingly, in the Discussion, we have added: “Importantly, only prenylated Gβγ dimer reaches the PM and is captured in our measurements of surface Gβγ, irrespective of total prenylated/non-prenylated cellular Gβγ content”.

2) Does the type of prenylation (e.g. farnesyl or geranylgeranyl) impact binding to the Gbetagamma anchor? It is not clear whether the anchor presents a specific binding pocket for

Gbetagamma that might be sensitive to different lipid modifications or combinations of Gbeta and Ggamma.

Addressing the role of the lipid moiety of Gy was out of the scope of this work. It is well established that Gy2 is geranylgeranylated (which is noted in the Results, 2nd paragraph), as most other Gy types, whereas Gy1 is farnesylated – see the review of Escriba et al. (2007). Interestingly, for several Gβγ interactors it has been shown that the type of prenylation is important for the specificity of interaction (mainly for PLCβ and ADCY; e.g. Myung et al. (2000), Fogg et al. (2001)), but not in many other cases (e.g. Jian et al. (2001), Lukov et al. (2004)). Unfortunately, owing to the limitation on text length, we could not discuss this any further (but we cite the relevant publications). Interestingly, as far as we know, the issue of farnesylation vs. geranylgeranylation has not been addressed for GIRKs, and future studies on this topic may provide insights into the structural details of Gy-GIRK interactions. However, physiologically this seems to be less important. The farnesylated Gy is specifically expressed in the retina, whereas there is no evidence that GIRKs play a role in this tissue.

Reply to Reviewer #2

Nature Communications 01/05/2025

Handklo-Jamal et al.

Live-cell quantitative monitoring reveals distinct, high-affinity Gβγ regulations of GIRK2 and GIRK1/2 channels

Summary: This manuscript reports the importance and function of Gy prenylation for affinity and subsequent activation of GIRK channels, and describes a Gβγ anchor located in the N-terminal and distal C-terminal region of G1 GIRK channels. The authors investigate these biochemical features of Gy and GIRK channels in *Xenopus* oocytes. This allows the authors to address a discrepancy in the field regarding Gβγ affinity for GIRK channels using prenylated or non-prenylated Gβγ. Assessing current in these *Xenopus* oocytes with prenylated and non-prenylated Gβγ shows significant affinity enhancements with prenylated Gβγ. The authors also utilize fluorophore tagged GIRK and Gy to obtain a Kd of Gβγ for GIRK 1/2 and GIRK2. They then define the Gβγ anchor on GIRK1 subunits via dotblot analysis and pull down quantitative western blotting to show deletion of G1-NT and G1-dCT reduces affinity for Gβγ. The authors also support their claims through molecular dynamic modeling that agrees with their biochemical conclusions.

This manuscript describes important biochemical interactions between Gβγ and GIRK channels that should be considered whenever investigating this interaction. However, there are some substantive issues that the authors should consider:

Major Critiques:

- In the section of the results titled “**G1-NT and G1-dCT form a Gβγ-binding site and contribute to channel’s interaction with Gy’s prenylation tail, Gy_{prenyl}”** claims are made that the prenylation of Gy is critical for the interaction of Gβγ with the G1-NT and G1-dCT fusion protein, as well as specifically the dCT, as its deletion causes less binding. While this was supported with MD simulations, this was not confirmed by a pull-down assay with non-prenylated Gβγ. If this claim is true, non-prenylated Gβγ should display a lower yield of G1NC and G2NC in that assay. While *in-*

silico methods are useful at identifying potential interactions, these should be confirmed *in-vitro* to make this claim.

We thank the reviewer for the insightful comment. While we have already demonstrated a significantly lower binding of full-length G1NC and G2NC to $G\beta\gamma_{C68S}$ compared to $G\beta\gamma_{WT}$ (Fig. 1 and Supplementary Fig. 2), we have not confirmed the prediction of the MD analysis that the fusion protein of G1-NT and G1-dCT, the G1NdCT construct, and most probably dCT alone, should also bind poorly to non-prenylated $G\beta\gamma$. We have conducted the proposed experiment and show that the prediction was correct for both G1NdCT and mCherry-fused G1-dCT (the new Supplementary Fig. 11). We have added the description of the experiments to Results (last paragraph: “In support of the important role of the NT-dCT unit for GIRK1 interaction with $G\gamma_{prenyl}$, we observed a complete loss of G1NdCT binding to $G\beta\gamma$ in the non-prenylated $G\gamma_{C68S}$ (Supplementary Fig. 11)”), and to Discussion, paragraph 10: “Importantly, *congruent with experimental results (Fig. 7, Supplementary Fig. 11)*, the simulations reveal a dynamically arising structural unit formed by G1-NT and G1-dCT...”

- Defining the $G\beta\gamma$ anchor within G1-NT and G1-dCT was done via truncation of the protein leading to a decrease in the $G\beta\gamma$ binding which is supported. However, truncation of channel subunits in this manner brings up concern about the biological implication of these deletions. Primary sequence length is known to be critical for proper protein folding. In this manuscript, there is no validation that the protein is in fact folded properly. Alanine mutation in these regions displaying the same results as the truncation would significantly strengthen the support of these claims.

We agree that using truncated channels may cause significant functional changes. However, we used constructs exactly corresponding to those used for structural studies, which was highly important for the understanding of the functional roles of the unstructured parts. The constructs used here have been extensively tested for function over the years in our and others' previous and this publication, including channels reconstituted in bilayers and expressed in heterologous cell models. The crystal and cry-EM structures present strong evidence for correct folding of the truncated channels. The distal terminal segments were removed in these works are mostly disordered, but their presence in the full-length channels would not destabilize the fold formed by the main structured elements.

Functionally, for GIRK2 with both NT and CT truncated as in crystal and cryo-EM structures of Mackinnon, Kurachi and Slesinger groups, response to agonist in cells (Whorton & Mackinnon 2011 DOI 10.1016/j.cell.2011.07.046) and to $G\beta\gamma$ in bilayers (Wang et al. 2014, 2016) and oocytes (Supplementary Fig. 5, in this MS) was preserved. Furthermore, here we show that the affinity to $G\beta\gamma$ in oocytes was not detectably affected (Supplementary Fig. 5), suggesting that the “ $G\beta\gamma$ -activation” site and the gating mechanism are preserved.

We have also extensively characterized the truncated GIRK1 (only dCT was removed) on whole-cell and single-channel level, in GIRK1/2 context (since GIRK1 does not express alone). We showed strong activation by agonist and $G\beta\gamma$ despite the reduction in basal activity; actually, the relative $G\beta\gamma$ -induced activation was even greater than in full-length channel, which we interpret to be the consequence of reduced $G\beta\gamma$ recruitment and I_{basal} that leaves more $G\beta\gamma$ -activation sites

unoccupied and available for additional occupation by expressed G $\beta\gamma$ (Kahanovitch et al. 2014). Therefore, we remain confident that the G $\beta\gamma$ -activation sites in this construct are intact.

We agree that alanine scanning or additional structure-function approaches (e.g. charge reversion mutations, hydrophobic vs. hydrophilic) can provide further details and insights into interaction surfaces and mechanisms. However, this is not trivial. We see in peptide mapping and in MD simulations that the G $\beta\gamma$ -binding regions are very large, and at present do not have any information about the importance of individual amino acids (although MD provides some interesting hints). Such detailed characterization will require a full new study which is beyond the scope of this MS.

Minor Critiques:

- In the first sentence of the third paragraph of the introduction, the G $\alpha\beta\gamma$ should be specified as G $\alpha_i/o\beta\gamma$.

Corrected.

Rply to Reviewer #4:

Handklo-Jamal et al. addressed a long-standing question in GIRK channel regulation by integrating peptide mapping with coarse-grained (CG) molecular dynamics (MD) simulations to propose a cooperative “G $\beta\gamma$ -anchor” mechanism involving the GIRK1 N-terminus, distal C-terminus, and geranylgeranyl tail of G γ . The combined use of structural modeling, simulations, and biochemical validation is both original and compelling. This work is of clear interest to the ion channel and signalling communities, particularly those studying GIRK regulation, GPCR–G protein–effector coupling, and the effects of lipid modification. This builds upon the foundational GIRK–G $\beta\gamma$ structures and expands our understanding of previously unresolved regulatory regions. Most conclusions are reasonably supported by the data; however, several important concerns remain:

1. All CG simulations were conducted without a membrane, despite the highly lipophilic nature of the geranylgeranyl moiety. In Supplementary Fig. S10, the tail frequently associates with GIRK surfaces that would ordinarily be embedded within the bilayer. To substantiate the proposed mechanism, the authors should either perform additional simulations incorporating a membrane (e.g. three CG replicas) or explicitly acknowledge this limitation and adjust their mechanistic claims accordingly.

We acknowledged this limitation in the previous version and thank the reviewer for urging us to perform this analysis. We added the membrane in three CG 5 μ s runs. G $\beta\gamma$ maintained strong interaction with G1NC. The main sites of interaction, in particular the G1NT, remained the same. Accordingly, we have changed Fig. 10d, to show two histograms of % of time spent by G1NC a.a. residues in contact with the G γ_{prenyl} , in simulations with and without the membrane. In addition, the new Supplementary Fig. 10c shows the overlap in the predicted GIRK1 segments that bind G γ_{prenyl} (including all-atom simulations). The full analysis of these simulations is included in Supplementary MD Figures Collection and Supplementary Table 14. Overall, the MD simulations suggest that, at least for the channel-bound G $\beta\gamma$, high-affinity G $\beta\gamma$ site in the G1 NT-CT scaffold

attracts prenylated G γ no less than the membrane, and G γ prenylated tail continuously interacts with GIRK1 even in the presence of the membrane. We further acknowledge that “Our simulations started with G $\beta\gamma$ pre-bound to the channel; fully *ab initio* simulations would require significantly longer sampling but could potentially reveal additional membrane interactions of G γ _{prenyl}. However, such analyses are beyond the scope of the current study” (Results, last section, last-but-one paragraph).

2. While CG simulations accelerate dynamics, a microsecond may still be insufficient to capture slower cooperative rearrangements, especially in multimeric membrane-associated systems. The authors should either extend their simulations to the multi-microsecond range or provide a more detailed analysis of the contact profiles across independent 1- μ s replicas, including evidence of convergence.

We have extended our simulations of the G1NC/G2NC systems and now have a total of 175 μ s of simulation time, including systems with G1NC/G $\beta\gamma$ embedded in a membrane and an all-atom simulation of G1NC. Compared to the 40 μ s of simulation time when the paper was first submitted, we have increased the total simulation time by about 400%. Importantly, there are no significant differences from the initial results obtained in our previous simulations for the interaction of G1NC and G2NC with G $\beta\gamma$. Additionally, we observed similar binding sites for the prenylation tail in both the G1NC system with a membrane and the all-atom simulations, as noted above and shown in the new Supplementary Fig. 10c, Supplementary Table 14, and Supplementary MD figures collection.

3. The reported interaction percentages and reductions are presented without uncertainties or significance testing. To strengthen their conclusions, the authors should report the mean \pm SEM across replicas and apply appropriate statistical tests when claiming significant differences between constructs.

We thank the reviewer for this suggestion and apologize for the omission. We have done that. The full set of data across replicas is shown in the new Supplementary Tables 12-14, which also report the results of statistical analyses of differences between different systems (G1NC vs, G2NC, full-length vs. truncated). Means \pm SEM, instead of just mean, are now presented in Fig. 8 and in the text.

4. To validate the stability and resolution of the proposed interaction beyond the CG level, the authors should perform at least one atomistic MD refinement of the interface.

We have performed 3 \times 0.5 μ s all-atom simulation runs for G1NC, with results very similar to the CG. The comparison is shown in the new Supplementary Fig. 10c. Full details of all the analysis of all simulations related to G γ _{prenyl}-channel and G1 NT-dCT interactions are presented in the new Supplementary Table 14.

5. The use of AlphaFold to model full-length GIRK subunits is an ambitious approach. However, the reliability of these predictions in multimeric complexes with flexible termini and membrane-associated regions should be discussed in greater depth, including any limitations and confidence levels of the predicted structures.

We have acknowledged this in the paper: “Despite the low-confidence of AlphaFold predictions for these elements, MD calculates interactions based on physical parameters and can capture dynamic interactions even if the initial structure is uncertain”. We have now added the experiment of Supplementary Fig. 11, showing a great decrease in G β γ interaction with G1NdCT (the “anchor” predicted by the MD and previous experiments) when G γ prenylation is absent. Accordingly, we emphasize that the MD predictions are supported by experiment: “Importantly, congruent with experimental results (Fig. 7, Supplementary Fig. 11), the simulations reveal a dynamically arising structural unit formed by G1-NT and G1-dCT... (Discussion, 11th paragraph).

We note parenthetically that, although the confidence of AlphaFold predictions for some of segments within the distal NT and CT was very low, on the average these segments were predicted with average or even high confidence except for the NT, as shown in the new Supplementary Table 11 (second section).

Reply to Reviewers.

We thank the reviewers once again for the constructive critique on the first version of this paper, which helped us to improve the paper, and for the unanimous acceptance of the revised version.